# Self-Generated In-Context Examples Improve LLM Agents for Sequential Decision-Making Tasks

**Vishnu Sarukkai**
Stanford University

**Zhiqiang Xie**
Stanford University

**Kayvon Fatahalian**
Stanford University

## Abstract

Improving Large Language Model (LLM) agents for sequential decision-making tasks typically requires extensive task-specific knowledge engineering—custom prompts, curated examples, and specialized observation/action spaces. We investigate a different approach where agents automatically improve by learning from their own successful experiences without human intervention. Our method constructs and refines a database of self-generated trajectories that serve as in-context examples for future tasks. Even naive accumulation of successful trajectories yields substantial performance gains across three diverse benchmarks: ALFWorld (73% to 89%), Wordcraft (55% to 64%), and InterCode-SQL (75% to 79%). These improvements exceed those achieved by upgrading from gpt-4o-mini to gpt-4o and match the performance of allowing multiple attempts per task. We further enhance this approach with two innovations: database-level curation using population-based training to propagate high-performing example collections, and exemplar-level curation that selectively retains trajectories based on their empirical utility as in-context examples. With these enhancements, our method achieves 93% success on ALFWorld—surpassing approaches that use more powerful LLMs and hand-crafted components. Our trajectory bootstrapping technique demonstrates that agents can autonomously improve through experience, offering a scalable alternative to labor-intensive knowledge engineering.

## 1 Introduction

When creating LLM agents for sequential decision-making tasks, practitioners often improve agent performance by investing in task-specific knowledge engineering (through tedious prompt tuning [1], human-crafted in-context examples [2, 3] or custom observation and action spaces [4, 5]). Using these techniques, scaling agent performance comes from scaling human effort.

In this paper, we investigate an alternative path: enabling LLM agents to autonomously bootstrap their own performance by leveraging their own successful experiences via in-context learning. The efficacy of in-context learning depends critically on both the quality of the examples [2, 3] and their relevance to the current decision point [6–8]. This insight provides a natural direction for automated agent self-improvement: accumulating successful self-generated trajectories and estimating the most relevant and effective prior experiences to use as in-context examples for each action.

Our work assumes a ReAct-style agent [9] that retrieves different examples for each decision point based on their relevance to the current situation [10, 11]. We build on this foundation by focusing specifically on how to construct and refine the underlying database of self-generated examples. How can we identify which trajectories enhance performance on new tasks versus those that hinder performance? This database construction problem requires addressing both the collection of high-quality trajectories and the strategic curation of the most valuable ones for future retrieval at each decision point in the agent's reasoning and acting loop.

39th Conference on Neural Information Processing Systems (NeurIPS 2025).

**Algorithm 1** ReAct-style Agent Loop

---

1: **function** AGENT$(g, \mathcal{D}, \mathcal{E}, T)$
2:      $C_p \leftarrow \texttt{Retrieve}(\mathcal{D}, keys = [g])$                               ▷ Retrieve for plan
3:      $p \leftarrow \text{LLM}_{\text{plan}}(g, C_p)$                                   ▷ Generate initial plan
4:      Initialize $\tau \leftarrow (g, p, \{\}, -)$; $o_1 \leftarrow \mathcal{E}.\texttt{obs}()$
5:      $C_1 \leftarrow \texttt{Retrieve}(\mathcal{D}, keys = [g, p, o_1])$                 ▷ Retrieve for current observation
6:      **for** $t = 1$ to $T$ **do**
7:          $r_t \leftarrow \text{LLM}_{\text{reason}}(\tau, o_t, C_t)$                          ▷ Generate reasoning
8:          $C_{t+1} \leftarrow \texttt{Retrieve}(\mathcal{D}, keys = [g, p, r_t])$        ▷ Retrieve for current reasoning
9:          $a_t \leftarrow \text{LLM}_{\text{act}}(\tau, o_t, r_t, C_{t+1})$                     ▷ Decide action
10:         $o_{t+1}, \text{done}, s \leftarrow \mathcal{E}.\texttt{step}(a_t)$                  ▷ Execute action in environment
11:         $\tau \leftarrow \tau \cup (o_t, r_t, a_t)$
12:         **if** done **then**
13:             **return** $(g, p, \{(o_i, r_i, a_i)\}_{i=1}^t, s)$
14:      **return** $(g, p, \{(o_i, r_i, a_i)\}_{i=1}^T, 0)$                    ▷ Failed due to timeout

---

We demonstrate that even naive database accumulation improves test-set performance from 73% to 89% on ALFWorld, 55% to 64% on Wordcraft, and 75% to 79% on InterCode-SQL. (Equivalent to what a baseline agent would achieve if it were allowed two to three attempts per task.) We further propose two database construction enhancements: (1) database-level curation that identifies and propagates high-performing example databases, and (2) exemplar-level curation that identifies helpful trajectories based on their empirical utility as in-context examples. These approaches do not require task-specific prompt engineering [12, 4] or custom observation/action space design [11, 4], but improve success rates on ALFWorld to 93%—surpassing approaches like AutoManual [4] that use more powerful LLMs and hand-crafted observation and action spaces, as well as hierarchical approaches like Autoguide [12]. The success rate improvement on ALFWorld exceeds the boost obtained from upgrading the agent's underlying LLM from gpt-4o-mini to gpt-4o. Our results highlight the practical value of trajectory bootstrapping as a dimension for scaling test-time compute.

## 2 Preliminaries

**Sequential Decision-Making Tasks** We focus on multi-step sequential decision-making tasks where agents must produce a series of actions based on observations of the environment. The sequential nature of these tasks introduces unique challenges for LLM agents, as they must interpret intermediate environmental feedback, maintain coherent reasoning across steps, and adapt their strategy based on the evolving task state.

We assume a standard POMDP setup (App. B), where an agent, given a task goal $g$, interacts with the environment $\mathcal{E}$ for up to $T$ timesteps. At each timestep $t$, the agent receives an observation $o_t$, takes an action $a_t$, and $\mathcal{E}$ transitions to the next state. We consider sparse-reward environments where success is only determined at the end of an episode. This is a standard setting in prior agentic work [9, 13, 12, 4]. Please see App. B for details.

**ReAct-style Agent Loop** Our work assumes a ReAct-style [9] agent architecture that employs recent best practices for in-context retrieval [10, 11]. The agent operates through a three-phase approach (planning, reasoning, and acting) as formalized in Alg. 1. Two key components differentiate our implementation from basic ReAct: (1) an initial planning step where the agent generates a high-level plan before execution begins ( 1, line 3), which has been shown to boost performance in prior work [10, 14, 13, 4], and (2) dynamic retrieval of different trajectory segments for each decision point [10, 11], rather than using the same examples throughout an episode [9, 13, 12].

The agent operates through three key LLM-based functions: LLM$_{\text{plan}}$ generates a high-level plan $p$ for the goal, LLM$_{\text{reason}}$ processes observations $o_t$ to produce reasoning $r_t$, and LLM$_{\text{act}}$ determines actions $a_t$. The $\texttt{Retrieve}()$ function selects the $k$ most relevant examples from database $\mathcal{D}$ based on similarity between lookup keys and examples. Our contribution focuses specifically on constructing and refining the trajectory database that powers this retrieval mechanism, without relying on task-specific

prompting, observation spaces [11] or action spaces [5, 4]. Note that in Alg. 1 all task-specific knowledge is encapsulated in $\mathcal{D}$. For simplicity, we eschew other techniques, like hierarchical learning [13, 12, 4], that are also task-agnostic, but add additional agent complexity.

## 3 Problem Statement

Given the ReAct-style agent described in Sec. 2, our goal is to construct a trajectory database that maximizes agent performance on sequential decision-making tasks. We focus on building and refining the example database accessed by the agent's retrieval mechanism. Formally, we aim to construct a trajectory database $\mathcal{D}$ where each trajectory $\tau \in \mathcal{D}$ captures a complete task attempt: $\tau = (g, p, \{(o_t, r_t, a_t)\}_{t=1}^T, s)$. We aim to maximize agent performance across tasks $\mathcal{T}$: $\mathcal{D}^* = \arg\max_{\mathcal{D}} \mathbb{E}_{g \sim \mathcal{T}}[\texttt{Success}(\texttt{Agent}(g, \mathcal{D}, \mathcal{E}, T))]$, where $\texttt{Success}()$ returns the binary outcome $s$. We assume that we are given: (1) $\mathcal{D}$ initialized with a small number of human-generated trajectories, (2) a descriptor of the action space, and (3) access to a set of training tasks drawn from $\mathcal{T}$ that the agent can attempt. All three assumptions are typical for ReAct-based agents [9, 10, 13, 12, 4].

## 4 Related Work

**In-context learning for agent improvement**    Despite the current popularity of reinforcement learning-based approaches for improving agent capabilities [15–18], in-context learning offers distinct scientific and practical advantages. These benefits include model-agnostic portability across different LLMs, efficiency in low-sample regimes [3, 19], and accessibility when implementation barriers exist for weight modification methods. Both empirical and theoretical work has established that in-context performance can scale effectively with additional examples [6, 7, 19, 8], suggesting that strategic example accumulation should lead to significant performance improvements. We focus on maximizing the value of limited examples through in-context methods, while hypothesizing that database quality, not just quantity, critically influences performance scaling. For completeness, we offer a preliminary investigation in App. F of how our collected trajectories could potentially serve as training data for fine-tuning approaches.

**Automatic in-context examples**    Recent work has demonstrated the effectiveness of optimizing both instructional content and example curation in prompts. DSPy [20] introduced a framework for optimizing multi-step pipelines through instruction tuning and strategic example curation. Self-generated examples containing reasoning traces can eliminate the need for human-written examples, and these self-generated examples often contribute more to performance than optimized instructions alone [21]. These approaches typically select fixed exemplars for all task instances, whereas our method enables the dynamic selection of different in-context examples for each decision.

**In-context self-improvement of LLM Agents**    Self-improvement methods for LLM agents either aim to solve one task (performing search/optimization) or transfer knowledge from prior tasks to novel ones (generalization) (see App. J.2 for further discussion). Approaches to solve a single task scale the number of sampled solutions at inference time [22–24] or incorporate feedback from failed attempts [25]. Knowledge transfer approaches include abstraction-based methods like ExpeL [13] and AutoGuide [12], while others employ task-specific information in their design—RAP [10] uses task-specific prompts and AutoManual [4] constructs task-specific state and action spaces (see App. J.1). Other dimensions of self-improvement include hierarchical execution [26] and optimization techniques for multi-stage systems[27–29]—techniques complementary to our approach. Rather than developing complex architectures or leveraging task-specific information, we focus on identifying which trajectories most contribute to successful outcomes as in-context examples.

**In-context reinforcement learning**    Our work connects to the emerging area of in-context reinforcement learning, where language models perform sequential decision-making through contextual examples rather than parameter updates. Recent work has explored how transformers can implement RL algorithms in-context, both via algorithm distillation for in-context RL [30], and via transformers' ability to learn from reward trajectory contexts via supervised pretraining [31]. LLMs can balance exploration-exploitation tradeoffs through intelligent prompt design [32], and other work suggests LLMs being able to perform in-context policy iteration [33]. While these approaches focus on learning RL algorithms or policies in-context, our work addresses a complementary problem: how to

---

**Algorithm 2** Database Curation Logic for +DB-Curation

---

1: **procedure** OPTIMIZEDATABASES($\{\mathcal{D}_1, \mathcal{D}_2, ..., \mathcal{D}_N\}$, $interval$)
2:     Initialize performance metrics $\{m_1, m_2, ..., m_N\}$ for each database
3:     **for** $t = 1, 2, ..., T_{train}$ **do**
4:         **for** $i = 1, 2, ..., N$ **in parallel do**
5:             Execute task $t$ using database $\mathcal{D}_i$
6:             If successful, add trajectory to $\mathcal{D}_i$
7:             Update rolling performance metric $m_i$ on recent tasks
8:         **if** $t = 10 \times 2^j$ for any $j \in \mathbb{N}$ **then**
9:             Sort databases by rolling performance on recent tasks
10:           Replace worst database with copies of best

---

effectively curate and retrieve trajectory examples to maximize in-context learning performance for sequential decision-making tasks.

## 5 Methods

We now discuss three algorithms for constructing database $\mathcal{D}$ using a continual collection approach.

### 5.1 *Traj-Bootstrap*: Constructing a Database of Previously-Solved Tasks

Our trajectory-bootstrapping algorithm Traj-Bootstrap constructs a trajectory database $\mathcal{D}$ by collecting successful agent experiences. As outlined in Sec. 3, we start with a minimal set of human-provided exemplars (which could be empty), then grow the database as the agent successfully completes training tasks. This process creates a positive feedback loop where successful examples help the agent solve new tasks, generating more successful examples.

Traj-Bootstrap operates on principles similar to reward-weighted regression in reinforcement learning [34], where only successful trajectories ($s = 1$) are stored in the database. This filtering mechanism ensures the agent learns from positive examples while avoiding potentially misleading failed attempts. Successful trajectories can be leveraged by asking the agent to imitate the successful patterns in these trajectories. Failed trajectories are more challenging to operationalize due to the credit attribution problem: it is necessary to identify the 'good' vs 'bad' parts of the trajectory so the the agent can imitate the good parts and avoid the mistakes made in the bad parts. Failed trajectories do offer the opportunity to guide exploration [25]; we leave this direction to future work.

### 5.2 *+DB-Curation*: Database-Level Data Curation

Traj-Bootstrap exhibits unpredictable performance variation across training (database construction) trials, even when following identical collection procedures. Fig. 1 illustrates this variation across five trials on the InterCode-SQL benchmark (a benchmark we use for evaluation in Sec. 6). The variance arises from two factors: (1) the stochasticity of LLM outputs creating different initial trajectories, and (2) an amplification effect where early differences in collected examples lead to wide performance variation.

This observation motivates a data curation strategy inspired by population-based training in reinforcement learning [35]. Fig. 1 shows that some databases lead to better task performance than others—so we identify the underperforming databases periodically during training and remove them, continuing growth from top-performing ones. We introduce *+DB-Curation*, a population-based training algorithm (Alg. 2) to identify and propagate the most effective databases during the bootstrapping process.

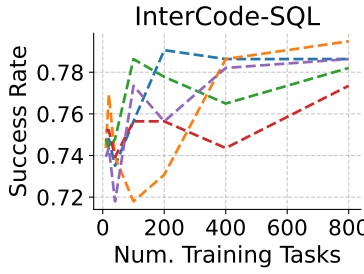

Figure 1: **Traj-Bootstrap exhibits variance in test-time success rate**. Dashed lines plot success rates achieved in five independent trials.

**Algorithm 3** Database Construction from Top Exemplars for +Exemplar-Curation

---
1: **procedure** SELECTEXEMPLARS($\{\mathcal{D}_1, \mathcal{D}_2, ..., \mathcal{D}_N\}, T_{train}$)
2:     $\mathcal{D}_{composite} \leftarrow \emptyset$
3:     Compute quality metric $Q(\tau)$ for each trajectory $\tau \in \bigcup_{i=1}^{N} \mathcal{D}_i$
4:     **for** each task $t \in T_{train}$ **do**
5:         $T_t \leftarrow \{$successful trajectories for task $t$ across all databases$\}$
6:         **if** $T_t$ is not empty **then**
7:             Select top-1 trajectory from $T_t$ by quality metric $Q$
8:             Add selected trajectory to $\mathcal{D}_{composite}$
9:     **return** $\mathcal{D}_{composite}$

---

+DB-Curation maintains $N$ database instances initialized with identical human-provided exemplars. Each instance is used by a separate agent that accumulates successful trajectories independently. Curation events occur when the number of tasks attempted reaches size thresholds (starting at size 10 and doubling thereafter). At each threshold, +DB-Curation evaluates database performance based on the agent's success rate on all training tasks since the last threshold, and replaces the worst-performing database with a copy of the top-performing database.

The key insight of this approach is that database quality emerges from collective properties—like coverage, diversity, and complementarity across examples—not just individual trajectory quality. Moreover, a single trajectory collected early in training can influence many future trajectories by guiding the agent toward particular solution strategies, creating cascading database-level effects. By selecting and propagating entire databases, we preserve these beneficial emergent properties while using a simple, computationally efficient evaluation metric based on recent performance.

### 5.3 *+Exemplar-Curation*: Exemplar-Level Data Curation

While database-level curation via +DB-Curation identifies entire sets of complementary trajectories, discarding whole databases can eliminate valuable trajectories. We find that even poor-performing databases contain individual high-quality trajectories that yield better outcomes when used as examples. Conversely, some successful trajectories may contain bad individual decisions that would be unhelpful to repeat. This observation motivates *+Exemplar-Curation*: identifying and selecting individual high-quality exemplars across multiple database instances based on their empirical utility as in-context examples. This approach parallels value-function learning in reinforcement learning [36], where we estimate the 'value' of each trajectory based on its contribution to successful outcomes.

We introduce a retrieval-weighted quality metric analogous to a value function to quantify each trajectory's contribution to successful outcomes:

$$Q(\tau) = \frac{\sum_{i \in \mathcal{R}(\tau)} o_i \cdot f_i(\tau)}{\sum_{i \in \mathcal{R}(\tau)} f_i(\tau)} \tag{1}$$

where $\mathcal{R}(\tau)$ is the set of tasks for which trajectory $\tau$ was retrieved, $o_i$ is the binary outcome of task $i$, and $f_i(\tau)$ is the retrieval frequency during task $i$.

This value metric measures how often a trajectory is associated with successful outcomes when retrieved as an in-context example. It weights outcomes by retrieval frequency, prioritizing trajectories frequently retrieved during successful completions while penalizing those associated with failures.

Alg. 3 outlines +Exemplar-Curation. For each task in the training set, it identifies all successful trajectories across all $N$ database instances and selects the exemplar with the highest value according to the metric. This approach constructs a composite database containing only the most effective exemplars as measured by their empirical contribution to successful outcomes on subsequent tasks.

### 5.4 Train-Time vs Test-Time LLM Costs

The curation methods (+DB-Curation and +Exemplar-Curation) maintain $N$ parallel database instances during training, and therefore require $N\times$ more LLM inference during training compared

to Traj-Bootstrap. However, all methods use the same quantity and distribution of training tasks. At test time, all three methods have identical computational costs–Alg. 1 is simply provided with a different database $\mathcal{D}$ for each method. This contrasts with approaches that scale the number of LLM calls per test-time task to improve performance [22–25]. Our methods shift computational burden to training while maintaining efficient inference, a property our in-context methods share with fine-tuning methods.

## 6 Experiments

We evaluate our database construction methods through experiments addressing three key questions:

- Database scaling: How does task success rate scale with increasing database size?
- Improving database construction: How much do population-based training and exemplar-level curation improve task success rate?
- Overall effectiveness: How do our approaches compare to alternative approaches leveraging task-specific domain knowledge or hierarchical algorithms?

### 6.1 Experimental Setup

#### 6.1.1 Benchmark Tasks

We evaluate our methods on three benchmarks: **ALFWorld** [37], a text-based environment for navigation and object manipulation; **InterCode-SQL** [38], an interactive coding environment for SQL query generation; and **Wordcraft** [39], a simplified adaptation of Little Alchemy requiring compositional reasoning to combine elements. These benchmarks were selected because they: (1) provide large enough task pools to support meaningful train/test splits, (2) represent diverse reasoning challenges relevant to sequential decision-making, and (3) have been used in prior work, enabling direct comparisons with existing methods.

#### 6.1.2 Methods Compared

Our methods include:

- **Fixed-DB**: The baseline agent as described in Sec. 2, with a fixed database of human-provided initial examples and no database growth.
- **Traj-Bootstrap**: The simple progressive accumulation approach from Sec. 5.1.
- **Traj-Bootstrap+DB-Curation**: Our database-level trajectory curation from Alg. 2.
- **Traj-Bootstrap+Exemplar-Curation**: Our exemplar-level trajectory curation from Alg. 3.
- **Traj-Bootstrap+DB+Exemplar-Curation**: Applying both our database-level trajectory curation and propagation and our exemplar-level trajectory curation.

We compare these methods to two hierarchical designs. **Autoguide** [12] converts successful trajectories into explicit rules and retrieves the most contextually relevant rules, alongside low-level trajectories, at inference time. **AutoManual** [4] leverages hand-crafted task-specific observation and action spaces–see App. J.1 for details. Unless otherwise specified, we use GPT-4o-mini as our base LLM. We report success rates averaged over five random seeds. See App. E for additional details.

### 6.2 Traj-Bootstrap Results

**Traj-Bootstrap performance improves with more training tasks** Tab. 1 presents the final success rate metrics for our database construction methods. The performance of Traj-Bootstrap generally improves with increases in the number of training tasks attempted (Fig. 2) Performance continues to improve with more training tasks across all benchmarks, but exhibits diminishing returns—most gains occur within the first 25% of added training tasks. This efficiency decline occurs because each new example is retrieved less frequently as the database grows, influencing fewer generations, a pattern consistent with findings from Bertsch et al. [19] and Agarwal et al. [8]. As mentioned in Sec. 5, we observe performance variability across trials and within individual trials. Cross-trial variance indicates that some trials produce higher-performing databases when solving identical tasks. Within-trial fluctuations show that certain added trajectories can degrade performance.

| Method | ALFWorld | InterCode-SQL | Wordcraft |
|---|---|---|---|
| Fixed-DB | 0.73±0.02 | 0.75±0.01 | 0.55±0.03 |
| Traj-Bootstrap | 0.89±0.01 | 0.79±0.01 | 0.64±0.03 |
| +DB-Curation | 0.91±0.01 | 0.78±0.01 | 0.64±0.01 |
| +Exemplar-Curation | 0.90±0.02 | 0.81±0.01 | **0.72**±0.02 |
| +DB+Exemplar-Curation | **0.93**±0.03 | **0.82**±0.01 | 0.69±0.01 |

Table 1: **Average success rate of our methods: self-collected trajectories provide the largest boosts in task success rate**. Traj-Bootstrap outperforms Fixed-DB across all three benchmarks. The combination of +DB-Curation and +Exemplar-Curation provides the best performance on both ALFWorld and InterCode-SQL. +Exemplar-Curation provides the best performance on Wordcraft.

| Method | LLM(s) | Num Training Tasks | ALFWorld |
|---|---|---|---|
| Autoguide [12] | gpt-3.5-turbo + gpt-4-turbo | 100 | 0.79* |
| Automanual [4] | gpt-4o-mini | 36 | 0.72±0.01 |
| | gpt-4-turbo + gpt-4o-mini | 36 | 0.91±0.01 |
| Fixed-DB | gpt-4o-mini | 0 | 0.73±0.05 |
| | gpt-4o | 0 | 0.88±0.02 |
| Traj-Bootstrap | gpt-4o-mini | 100 | 0.84±0.04 |
| | gpt-4o-mini | 3500 | 0.89±0.01 |
| +DB-Curation | gpt-4o-mini | 100 | 0.86±0.02 |
| | gpt-4o-mini | 3500 | 0.91±0.01 |
| +Exemplar-Curation | gpt-4o-mini | 100 | 0.86±0.03 |
| | gpt-4o-mini | 3500 | 0.90±0.02 |
| +DB-Curation +Exemplar-Curation | gpt-4o-mini | 100 | 0.81±0.02 |
| | gpt-4o-mini | 3500 | **0.93**±0.03 |

Table 2: **Comparison of agent success rates on ALFWorld: contextualizing the performance of Traj-Boostrap.** The 15-point boost in average success rate from database construction via Traj-Bootstrap is similar to that achieved from upgrading Fixed-DB from gpt-4o-mini to gpt-4o. The performance of Traj-Bootstrap+DB+Exemplar-Curation exceeds Automanual [4], even though Automanual utilizes hand-designed observation and action spaces and a better LLM (gpt-4-turbo+gpt-4o-mini). * indicates results reported from original papers.

**+DB-Curation boosts performance on ALFWorld**  Fig. 3 illustrates how +DB-Curation can improve upon Traj-Bootstrap's performance, despite exhibiting occasional performance dips at smaller database sizes. These dips result from inaccurate estimates (due to low sample count) of database quality early in the process—introducing noise into the curation process.

**+Exemplar-Curation boosts performance on InterCode-SQL and Wordcraft**  As seen in Tab. 1, +Exemplar-Curation yields improvements in final task success rates on InterCode-SQL and Wordcraft, and also boosts success rate at intermediate database sizes for both InterCode-SQL and Wordcraft (Fig.3). To further highlight the impact of our exemplar-level curation metric, Fig. 4 compares databases built from the 'best' trajectories that are the most empirically effective in-context examples versus the least effective trajectories, as determined by Equation 1 in Sec. 5.3. The 'best' curve is identical to +Exemplar-Curation, while the 'worst' curve selects the bottom-1 trajectory instead of top-1 in Alg. 3, line 7. Using the database of high-quality examples yields a higher success rate across all database sizes for ALFWorld and Wordcraft, and for smaller database sizes for InterCode-SQL.

**+DB+Exemplar-Curation achieves best performance on ALFWorld and InterCode-SQL**  Fig. 3 and Tab. 1 highlight that +DB-Curation and +Exemplar-Curation can be complementary, as the combined +DB+Exemplar-Curation achieves the best final task success rates on both ALFWorld and InterCode-SQL (0.93 and 0.82 respectively). Note that on Wordcraft, +DB-Curation fails to

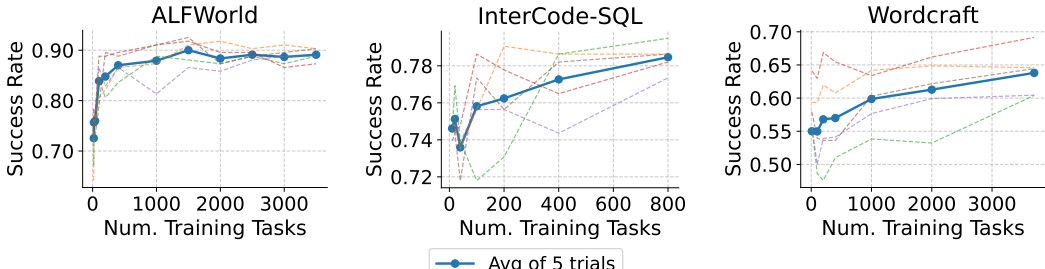

Figure 2: **Traj-Bootstrap results: success rate improves with increasing training tasks on all three benchmarks**. Individual trials (5) shown as dashed lines. All benchmarks exhibit diminishing returns as the database size increases. Trials show substantial performance variability, both within individual trials and across different trials.

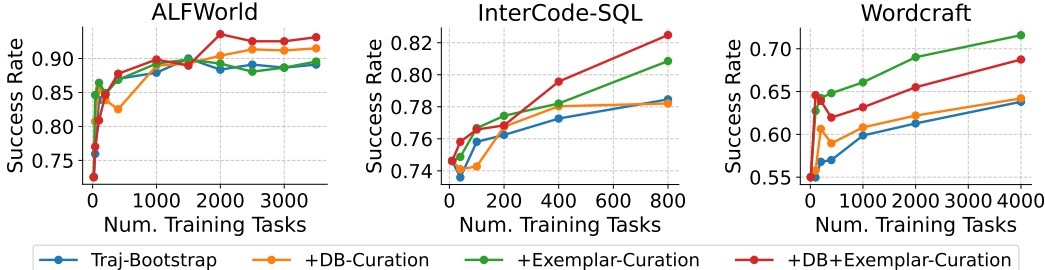

Figure 3: **Success rate comparison for Traj-Bootstrap and its variants (+DB-Curation, +Exemplar-Curation, +DB+Exemplar-Curation).** +DB-Curation enhances final success rate only on ALFWorld, but improves success rate for smaller DB sizes on all benchmarks. +Exemplar-Curation delivers success rate gains on both Intercode-SQL and Wordcraft. The combination of both enhancements delivers the largest gains on both ALFWorld and InterCode-SQL.

provide boosts whether or not +Exemplar-Curation is used–Traj-Bootstrap and +DB-Curation perform identically (0.64), and +Exemplar-Curation(0.71) outperforms +DB+Exemplar-Curation (0.67).

### 6.3 Contextualizing performance boosts from Traj-Bootstrap

To contextualize the improvements achieved by Traj-Bootstrap, we compare with several alternative strategies: test-time sampling, using a better LLM, task-specific strategies, and hierarchical strategies.

**Comparison with test-time scaling**   Our trajectory bootstrapping approaches achieve success rate improvements equivalent to scaling test-time compute by making multiple task attempts—an advantage in scenarios where multiple attempts are impractical or when success verification is unavailable at test time. Furthermore, our approaches provide these benefits without requiring any modifications to the test-time inference process. To demonstrate the magnitude of this benefit, we compare our methods to the alternative strategy of making multiple attempts at each test task with the Fixed-DB baseline and selecting the best outcome [22–24]. Tab. 3 reports the pass@k metrics for Fixed-DB across all three benchmarks, representing the probability of at least one successful attempt when making $k$ independent attempts at each task. Using only a single attempt per task, Traj-Bootstrap approach achieves success rate comparable to Fixed-DB pass@2 or pass@3 on all three benchmarks. +DB-Curation and/or +Exemplar-Curation perform nearly on the level of Fixed-DB pass@4 on ALFWorld, pass@5 on InterCode-SQL, and pass@5 for Wordcraft.

**Comparsion with model upgrades**   On ALFWorld, after 3500 training tasks Traj-Bootstrap yields a 20-point success rate boost over Fixed-DB, significantly outperforming the 15-point improvement gained by upgrading Fixed-DB to a more powerful LLM.

**Comparison with task-specific strategies**   Tab. 2 shows that Traj-Bootstrap+DB+Exemplar-Curation using GPT-4o-mini achieves a success rate exceeds (0.93) that exceeds that of Automanual [4] configured to use a combination of GPT-4-turbo and GPT-4o-mini (0.91). Thus, our methods

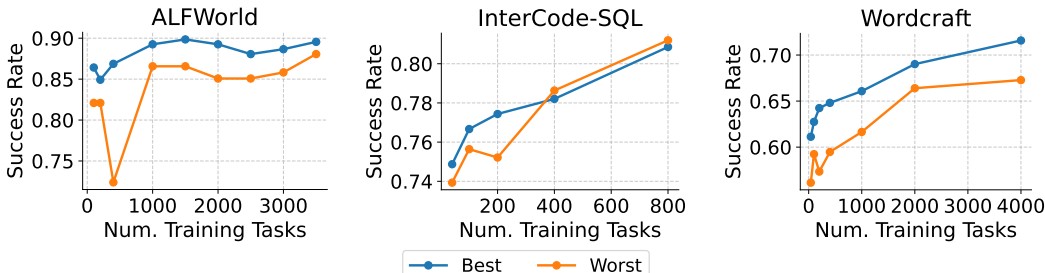

Figure 4: **The 'best' bootstrapped trajectories compared to the 'worst'.** Databases constructed from the highest-quality successful trajectory per task, as measured by Eq. 1, outperform databases built from the lowest-quality successful trajectories on both ALFWorld and Wordcraft. The 'best' curve is identical to +Exemplar-Curation, while the 'worst' curve selects the bottom-1 trajectory instead of top-1 in Alg. 3, line 7.

| Method | ALFWorld | Intercode-SQL | Wordcraft |
|---|---|---|---|
| Traj-Bootstrap | 0.89±0.01 | 0.79±0.01 | 0.64±0.03 |
| +DB+Exemplar-Curation | 0.93±0.03 | 0.82±0.01 | 0.69±0.01 |
| Fixed-DB@1 | 0.73±0.03 | 0.75±0.01 | 0.55±0.03 |
| Fixed-DB@2 | 0.87±0.02 | 0.78±0.03 | 0.62±0.02 |
| Fixed-DB@3 | 0.92±0.02 | 0.80±0.03 | 0.64±0.02 |
| Fixed-DB@4 | 0.94±0.02 | 0.81±0.02 | 0.66±0.02 |
| Fixed-DB@5 | 0.96±0.02 | 0.82±0.03 | 0.72±0.02 |

Table 3: **Pass@k of Fixed-DB on all benchmarks.** On all benchmarks, using only a single test-time attempt per task, Traj-Bootstrap achieves success rates between that of Fixed-DB at pass@2 and at pass@3. +DB+Exemplar-Curation achieves success rates between pass@3 and pass@5.

outperform an approach that uses a more powerful LLM and customized observation and action spaces. See App. K for a comparison to hand-crafted approaches on InterCode-SQL.

**Comparison with hierarchical algorithms**   Given 100 training tasks, Autoguide, a hierarchical rule-learning approach, achieves a 0.79 success rate (using a combination of gpt-3.5-turbo + gpt-4-turbo). Given the same number of training tasks our best approach achieves significantly greater success rate (0.86) with gpt-4o-mini (Tab. 2). While this comparison employs different algorithms and LLMs, the performance of Traj-Bootstrap suggests that self-constructed databases of low-level trajectories can be competitive with hierarchical approaches.

### 6.4   Extending Traj-Bootstrap

**Can we predict agent success?**   Beyond improving agent performance, we can also utilize our self-collected examples to implement useful agent diagnostics, such as predicting an agent's success on novel tasks. On InterCode-SQL and Wordcraft, we train a calibrated Random Forest classifier of agent success based on task goal and initial observation embeddings. Classifier quality (measured via AUROC) improves with database size, reaching 0.77 for InterCode-SQL and 0.71 for Wordcraft with our largest databases. The predicted probabilities also closely match empirical success rates, indicating well-calibrated predictions. See App. H for details.

**Can we use our self-collected databases for fine-tuning?**   We fine-tune GPT-4o-mini using trajectories from our best-performing Traj-Bootstrap+DB+Exemplar-Curation database for each benchmark. The resulting fine-tuned agents (ReAct-Finetune) outperform our in-context approach on ALFWorld (23-point vs. 20-point boost) and Wordcraft (19-point vs. 14-point), but perform worse on InterCode-SQL (4-point vs. 7-point). This suggests our self-collected examples are effective not only for in-context learning but also for creating fine-tuned agents. See App. F for details.

**Does our approach generalize across models?**   To test whether our method captures fundamental task structure rather than model-specific artifacts, we evaluate Mixtral 8x7B Instruct v0.1 using

databases collected with GPT-4o-mini. Despite this challenging cross-model transfer scenario, our full method achieves substantial improvements: +28 points on ALFWorld (0.55 vs. 0.27), +18 points on IC-SQL (0.70 vs. 0.52), and +12 points on Wordcraft (0.52 vs. 0.40). These gains match or exceed those observed with GPT-4o-mini itself, demonstrating that curated databases can effectively transfer across different model architectures and families. See App. G for details.

# 7  Discussion

**Contextualizing the costs of scaling performance along various axes**   Our self-improvement algorithm offers substantial cost savings compared to two common approaches for scaling model performance: (i) switching to a larger LLM, and (ii) test-time scaling via multiple attempts per task (assuming access to a perfect verifier).

Our full configuration with 5 parallel databases requires 3,500 × 5 training trajectories on GPT-4o-mini, with a worst-case database construction cost of $600. After this one-time investment, the per-task inference cost is only $0.034 (see Appendix D for detailed token usage and cost calculations). As shown in Section 6.3, this approach achieves a 0.93 task success rate on ALFWorld—outperforming GPT-4o with Fixed-DB (0.88) and matching GPT-4o-mini with Fixed-DB and 3 attempts per task (0.92).

In comparison, using GPT-4o at inference time costs $0.57 per task, breaking even with our approach after only  1,100 tasks. Test-time scaling with 3 attempts costs $0.10 per task, breaking even after 8,750 tasks. Beyond these thresholds, our method remains consistently more cost-effective while delivering superior performance.

At production scale, these savings become substantial. Consider a service processing one million ALFWorld tasks daily—a modest industry-scale throughput. Our method would save $530,000 daily versus GPT-4o, or $60,000 daily versus GPT-4o-mini with test-time compute, even accounting for all offline construction costs.

**Failed trajectories are used to improve the system**   During the initial database construction step, our method achieves train-set success rates of 81.3% (ALFWorld), 76.9% (InterCode), and 58.6% (Wordcraft) on average across 5 training runs. This means 18.7%, 23.1%, and 41.4% of trajectories are discarded respectively. While failed trajectories are not retained for in-context use at test time, failed trajectories are used to improve agent performance via our Exemplar-Curation algorithm (Section  5.3). We do not provide the failed trajectories directly in-context due to the challenges of credit attribution—given a long trajectory with a single incorrect action, it is challenging for the LLM to identify the actions within that trajectory to emulate and the ones to avoid repeating.

**Future directions**   The success of our approach reveals performance gains that stem primarily from accumulating successful examples, establishing a foundation for agent self-improvement where the quantity and quality of accessible data rivals the importance of architectural complexity. This parallels trends in traditional deep learning, where data curation often yields substantial improvements. Our findings point to promising research directions that approach LLM agent enhancement from a data-centric perspective—advancing both strategic data collection methods (balancing exploration versus exploitation across diverse tasks) and refined filtering techniques to maximize performance.

**Acknowledgments**   Thank you to Brennan Shacklett, Purvi Goel, Zander Majercik, William Wang, Bradley Brown, Jon Saad-Falcon, and William Mark for valuable discussions and feedback. This work was supported by the Stanford HAI SEAMS program, Roblox, and Meta, and API credits were provided by OpenAI and together.ai.

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

## A    Limitations and Broader Impacts

We make the assumption that we are given a few human-provided examples at the start of the database construction process–an assumption standard in the Agentic literature [9, 13, 12, 4, 10]. In App. I, we explore the alternate setting of starting from an empty database, and look forward to further research on bootstrapping from zero human examples in the future. Our algorithm is dependent on the in-context capabilities of LLMs–so our algorithms may be less effective if using an LLM with weaker in-context capabilities than gpt-4o-mini, but may be more effective if paired with more advanced LLMs. In addition, our algorithm generally produces improvements in task success rates, but task success rates are not monotonically increasing, and we hope future work will help improve both the monotonicity and sample efficiency of our algorithm.

This work has the potential to allow LLM Agents applied to a variety of task domains to self-improve. This provides a variety of benefits in terms of task performance, but could lead to applications in the future where the agents perform reward hacking-type behavior–performing undesirable behaviors that are not captured in task success/failure. Since the behavior of our agents is controlled via in-context examples, a potential mitigation technique could be to manually inspect the databases of self-generated examples, or even use an LLM Judge to inspect the examples.

## B    Note on Sequential Decision-Making Tasks

We focus on multi-step sequential decision-making tasks where agents must produce a series of actions over time based on observations of the environment. The sequential nature of these tasks introduces unique challenges for LLM agents, as they must interpret intermediate environmental feedback, maintain coherent reasoning across multiple steps, and adapt their strategy based on the evolving task state. This contrasts with one-shot generation tasks (e.g., solving math problems [40], one-shot code generation [41]) where feedback is only available after the complete solution is provided. Our example-driven learning strategy is potentially also suitable to single-step decision-making tasks, but we focus on the multi-step setting due to its applicability to a number of agentic tasks in real-world settings (embodied agents [14], browser-based tasks [42], etc.).

Formally, these tasks can be modeled as Partially Observable Markov Decision Processes (POMDPs), represented by the tuple $(\mathcal{S}, \mathcal{O}, \mathcal{A}, \mathcal{T}, \mathcal{R}, \gamma)$, where $\mathcal{S}$ denotes the underlying state space, $\mathcal{O}$ the observation space, $\mathcal{A}$ the action space, $\mathcal{T} : \mathcal{S} \times \mathcal{A} \to \mathcal{S}$ defines the deterministic transition function, $\mathcal{R} : \mathcal{S} \times \mathcal{A} \to \mathbb{R}$ is the reward function, and $\gamma \in [0, 1]$ is the discount factor. The partial observability reflects that agents don't have direct access to the full environment state but rather receive observations that provide limited information.

Given a task goal $g$, an episode consists of the agent interacting with the environment for a maximum of $T$ timesteps. At each timestep $t$, the agent receives an observation $o_t \in \mathcal{O}$ of the current state, takes an action $a_t \in \mathcal{A}$, and the environment transitions to the next state according to the transition function $\mathcal{T}$. In our setting, we specifically consider sparse-reward environments where success is only determined at the end of an episode—the agent receives $\mathcal{R} = 1$ for successful task completion and $\mathcal{R} = 0$ otherwise. This is a standard setting in prior agentic work [9, 13, 12, 4].

## C    Key Agent Details

In Sec. 2, we establish an agent design that enables it to learn in-context from its own self-collected experiences. Here, we elaborate on a few key design decisions in our agent design:

- **Standardized prompts**: we use the same simple, task-agnostic prompt templates for all tasks, rather than writing new prompts per task. These prompts are in App. E. Alternate appraoches incorporate domain-specific information into their prompts–we discuss these approaches in Appendices J.1 and  K.

- **Two-level retrieval**: We retrieve trajectories at both trajectory level (for planning) and state level (for reasoning and action selection), enabling the agent to leverage both strategic patterns and situation-specific techniques. Database $\mathcal{D}$ contains self-collected trajectories, and retrieval is performed at the trajectory level for the initial plan $p$, and in the state-level observation-reasoning-action loop for both $r_t$ and $a_t$.

---
**Algorithm 4** Multi-key Retrieval
---
1: **procedure** MULTIKEYRETRIEVAL($\mathcal{D}$, traj_keys, state_key, query, $k$, window_size)
2:     similarities $\leftarrow$ []
3:     **for** each trajectory $\tau$ in $\mathcal{D}$ **do**
4:         sim $\leftarrow 0$
5:         **for** each key in traj_keys **do**
6:             sim $\leftarrow$ sim + CosineSimilarity(query[key], $\tau$[key])
7:         sim $\leftarrow$ sim$/|$traj_keys$|$              $\triangleright$ Average similarity across trajectory keys
8:         similarities.append(sim, $\tau$)
9:     similar_trajectories $\leftarrow$ TopK(similarities, $k$)
10:    **if** state_key is not None **then**               $\triangleright$ State-level retrieval with window
11:        windowed_results $\leftarrow$ []
12:        **for** each trajectory $\tau$ in similar_trajectories **do**
13:            state_similarities $\leftarrow$ []
14:            **for** each state $s$ in $\tau$.states **do**
15:                state_sim $\leftarrow$ CosineSimilarity(query[state_key], $s$[state_key])
16:                state_similarities.append(state_sim, $s$, index($s$))
17:            _, most_similar_state, idx $\leftarrow$ Max(state_similarities)
18:            window_start $\leftarrow \max(0, \text{idx} - \lfloor \text{window\_size}/2 \rfloor)$
19:            window_end $\leftarrow \min(|\tau.\text{states}|, \text{idx} + \lceil \text{window\_size}/2 \rceil)$
20:            windowed_results.append($\tau$.states[window_start : window_end])
21:        **return** windowed_results
22:    **else**
23:        **return** similar_trajectories
---

- **Multi-key retrieval**: All retrieval is performed by KNN, with similarity metric defined as the average of cosine similarities across the specified 'key' variables. For instance, in Line 3 of Alg. 1, we retrieve from $\mathcal{D}$ using two keys: goal $g$ and plan $p$. We return similar trajectories based off the average of the cosine similarities of goals and plans when comparing each trajectory to the current trajectory. When doing state-level retrieval (Lines 7 and 10), we additionally find the most similar states within the selected trajectories via state-level key $o_t$ or $r_t$, then return a window of states around the most similar state. This is similar to the retrieval scheme in [10]. See detailed pseudocode for retrieval in Alg. 4.

- **Thought-based retrieval**: For the first step of a trajectory, we retrieve using the trajectory-level keys $(g,p)$ as well as the current observation $o_1$ (Alg. 1, line 6)–but for all subsequent steps we use reasoning $r_t$ as a key instead of observation $o_t$ (Alg. 1, line 9). This approach, inspired by Zhou et al. [11], enables generalization across trajectories with similar reasoning, and similarity across natural-language $r_t$ can be handled by generic embedding functions more easily than potentially bespoke observations $o_t$. By retrieving at every step, we aim to retrieve the most relevant trajectories for each decision.

- **Generic embedding mechanism**: Since $g$, $p$, and $r_t$ are all natural-language strings, we employ standard embeddings (all-MiniLM-L6-v2 [43]) that generalize across domains without task-specific engineering.

# D   Token Usage and Cost Analysis

We provide a detailed breakdown of token usage and costs for our self-improvement approach across all three benchmarks. This analysis demonstrates the economic viability of our method compared to alternative approaches for scaling model performance.

## D.1   Per-Episode Token Usage

Table 4 summarizes the average token consumption per episode across our three evaluation domains. Token counts reflect the full agent execution trace, including both reasoning and action generation steps.

| Benchmark | Avg Input Tokens | Avg Output Tokens | Max Episode Length |
|-----------|------------------|-------------------|--------------------|
| Wordcraft | 5,047 | 68 | 8 requests |
| InterCode | 5,385 | 37 | 20 requests |
| ALFWorld | 3,706 | 30 | 61 requests |

Table 4: Average token usage per episode across benchmarks. Episode length refers to the maximum number of API requests needed to complete a task.

The token usage pattern reflects our agent architecture design. For any task, we require **2 requests per step** (one for reasoning, one for acting), plus **1 additional request** when using planning at the episode start. The variance in input token counts primarily reflects differences in accumulated trajectory history within an episode.

### D.2 Per-Episode Cost Breakdown

Using GPT-4o-mini pricing (as of writing: $0.15 per 1M input tokens, $0.60 per 1M output tokens), we calculate the cost per episode for each benchmark:

**Wordcraft:** $\text{Cost} = (5{,}047 \times 0.15 + 68 \times 0.60)/1{,}000{,}000 \times 8 \approx \$0.006$ per episode

**InterCode:** $\text{Cost} = (5{,}385 \times 0.15 + 37 \times 0.60)/1{,}000{,}000 \times 20 \approx \$0.016$ per episode

**ALFWorld:** $\text{Cost} = (3{,}706 \times 0.15 + 30 \times 0.60)/1{,}000{,}000 \times 61 \approx \$0.034$ per episode

### D.3 Training Database Construction Costs

Our full self-improvement configuration uses 5 parallel databases with trajectory bootstrapping and exemplar curation. For ALFWorld, we collect 3,500 training trajectories per database; for Wordcraft, we collect 4,000 trajectories per database; and for InterCode, we collect 800 trajectories per database. The worst-case cost for database construction is:

| Benchmark | Episodes per DB | Total Episodes | Total Training Cost | Per-Task Inference Cost |
|-----------|-----------------|----------------|---------------------|-------------------------|
| Wordcraft | 4,000 | 20,000 | $120 | $0.006 |
| InterCode | 800 | 4,000 | $64 | $0.016 |
| ALFWorld | 3,500 | 17,500 | $595 | $0.034 |

Table 5: Training and inference costs across benchmarks. Total episodes calculated as episodes per database × 5 parallel databases.

Note that these costs represent **one-time upfront investments**. Once the databases are constructed, the per-task inference cost remains constant regardless of how many test-time tasks are processed.

### D.4 Cost Comparison with Baseline Approaches

We compare our approach against two common alternatives:

**Alternative 1: Using a larger model (GPT-4o).** At current pricing ($5.00/$15.00 per 1M input/output tokens), GPT-4o costs approximately:

- Wordcraft: $0.20 per task
- InterCode: $0.54 per task
- ALFWorld: $0.57 per task

**Alternative 2: Test-time scaling with multiple attempts.** Using GPT-4o-mini with 3 attempts per task (assuming a perfect verifier):

- Wordcraft: $0.018 per task
- InterCode: $0.048 per task
- ALFWorld: $0.102 per task

## D.5 Breakeven Analysis

For ALFWorld (our most expensive benchmark), our method breaks even with:

- **GPT-4o** after $\$595/(\$0.57 - \$0.034) \approx 1{,}110$ test tasks
- **GPT-4o-mini with 3 attempts** after $\$595/(\$0.102 - \$0.034) \approx 8{,}750$ test tasks

## D.6 Production-Scale Cost Savings

To contextualize these savings at production scale, consider a hypothetical agentic service processing 1 million ALFWorld tasks daily (a modest industry-scale throughput):

**Daily cost comparison:**

- Our approach: $1\text{M} \times \$0.034 + \$595$ (amortized) $\approx \$34{,}595$
- GPT-4o baseline: $1\text{M} \times \$0.57 = \$570{,}000$
- GPT-4o-mini with 3 attempts: $1\text{M} \times \$0.102 = \$102{,}000$

**Daily savings:**

- vs. GPT-4o: $535,405 ($195M annually)
- vs. 3-attempt baseline: $67,405 ($24.6M annually)

These calculations demonstrate that even modest upfront investments in self-improvement can yield substantial returns at production scale, while simultaneously improving task performance.

# E  Additional Implementation Details

## E.1 Hyperparameters

Unless otherwise specified, we use GPT-4o-mini as our base LLM (temperature 0.1). For Fixed-DB and all Traj-Bootstrap agents, we retrieve the top-$k$ most similar trajectories at each decision step ($k = 6$ for ALFWorld and InterCode-SQL, 10 for Wordcraft). We initialize each database with a small human-provided example set (18 for ALFWorld, 10 for InterCode-SQL, 4 for Wordcraft). With +DB-Curation, we maintain $N = 5$ database instances with curation every time the database size is doubled, starting with a minimum size of ten trajectories. We report success rates averaged over five random seeds. The standard deviation of the success rates is also reported. By default, we report success rate given the database at the end of the training process.

## E.2 Prompt Templates

Across all benchmarks, we use standardized prompt templates for the core components of our retrieval-based ReAct agent. The same templates were used across all benchmarks with no task-specific modifications. These templates are intentionally minimalist, focusing on providing the necessary context and retrieved examples while avoiding task-specific prompt engineering.

The templates are included below. Across all templates, the in-context examples follow the format specified in the prompt itself (for plan, the in-context examples are of form "goal,plan", etc):

Plan:

```
1 system_prompt: 'You are an expert at generating high-level plans of actions to
  ↪    achieve a goal.\n Here is your action space: {action_space}.\n Here are
  ↪    some examples of goal,plan from episodes that successfully achieved
  ↪    similar goals: {examples}'
2 user_prompt: 'goal: {goal}\n plan: '
```

Reason:

```
1 system_prompt: 'You are an expert at reasoning about the most appropriate
  ↪    action to take towards achieving a goal.\n Here is your action space:
  ↪    {action_space}.\n Here are some examples of
  ↪    goal,plan,observation,reasoning,action from episodes that successfully
  ↪    achieved similar goals: {examples}'
2 user_prompt: 'goal: {goal}\n plan: {plan}\n trajectory: {trajectory}\n
  ↪    reasoning: '
```

Act:

```
1 system_prompt: 'You are an agent in an environment. Given the current
  ↪    observation, you must select an action to take towards achieving the goal:
  ↪    {self.goal}.\n Here is your action space: {action_space}.\n Here are some
  ↪    examples of goal,plan,observation,reasoning,action from episodes that
  ↪    successfully achieved similar goals: {examples}'
2 user_prompt: 'goal: {goal}\n plan: {plan}\n trajectory: {trajectory}\n action:
  ↪    '
```

### E.3  Retrieval Implementation

For all retrieval steps, we implement hybrid search across all the desired retrieval keys–ex. goal, plan, observation, reasoning. We return the top-$k$ examples by averaged distance across each of the keys. We implement a sliding window approach for state-level retrieval to enhance contextual relevance–we include the surrounding context (preceding and following states) up to a window of 5 steps to provide coherent episode fragments.

The retrieval mechanism is implemented using FAISS [44] for efficient similarity search as the database grows. We use exact nearest neighbor search.

### E.4  Population-Based Training Details

Our database-level curation approach maintains a population of 5 database instances. Each instance is initialized with the same set of human-provided exemplars. The population undergoes curation every time the database size doubles, and performance is evaluated on the tasks attempted since the previous doubling.

The replacement strategy follows standard population-based training practices: the bottom 20% of databases (based on validation performance) are replaced with copies of the top 20%.

### E.5  Quality Metric Computation

For exemplar-level curation, we track the retrieval patterns of each trajectory throughout the training process. For each task, we record: 1. Which trajectories were retrieved 2. How many times each trajectory was retrieved during the solution process 3. Whether the task was successfully completed

After completing all training tasks, we compute the quality metric $Q(\tau)$ for each trajectory $\tau$ as:

$$Q(\tau) = \frac{\sum_{i \in \mathcal{R}(\tau)} o_i \cdot f_i(\tau)}{\sum_{i \in \mathcal{R}(\tau)} f_i(\tau)} \tag{2}$$

where $\mathcal{R}(\tau)$ is the set of tasks for which trajectory $\tau$ was retrieved, $o_i \in \{0, 1\}$ is the outcome of task $i$, and $f_i(\tau)$ is the retrieval frequency of trajectory $\tau$ during task $i$.

| Method | ALFWorld | InterCode-SQL | Wordcraft |
|---|---|---|---|
| Traj-Bootstrap+DB+Exemplar-Curation | 0.93±0.03 | **0.82**±0.01 | 0.69±0.01 |
| ReAct-Finetune | **0.96**±0.01 | 0.79±0.01 | **0.74**±0.01 |

Table 6: **Trained on the same data, fine-tuned agent ReAct-Finetune is competitive with our best in-context approach.** This suggests that our self-collected data is effective not only for in-context prompting but also for creating fine-tuned agents. All values are averages over 5 trials.

To ensure statistical significance, we only compute the quality metric for trajectories that were retrieved for at least 3 different tasks. For trajectories with insufficient retrieval data, we assign a neutral quality score equal to the average success rate across all tasks.

### E.6 Note on planning step

Following the convention from RAP [10], we omit the planning step on benchmarks with short trajectory length (Intercode-SQL, Wordcraft). This planning step is valuable for maintaining long-horizon coherence on the ALFWorld benchmarks (30 steps), and is standard in prior ReAct-based agentic work [10, 13, 12], whether the planning step is explicitly separate from reasoning, or incorporated into the first reasoning step.

## F Can Self-Collected Examples Improve a Fine-Tuned LLM Agent?

We have shown that self-collected databases improve the performance of in-context LLM agents. Here, we test whether the same data can also benefit fine-tuning.

Using the OpenAI fine-tuning API, we fine-tune GPT-4o-mini on each benchmark using data from our best-performing database construction method: Traj-Bootstrap+DB+Exemplar-Curation, collected over the full training set. We fine-tune using a simple ReAct-format prompt:

```
1  {
2  'system':'You are a ReAct agent that helps users accomplish tasks. Given a
   ↪  goal, you will receive observations about the environment and respond with
   ↪  your reasoning and actions. For each observation, first think through the
   ↪  problem step by step (Thought), then decide on an action (Action). Your
   ↪  actions should be clear, concise, and directly executable in the
   ↪  environment.',
3  'user':'Goal: {goal} \n Initial observation: {observations[0]}',
4  'assistant':'Thought: {reasoning[i]}\nAction: {action[i]}',
5  'user':'Observation: {observations[i+1]}',
6  ...
7  }
```

We refer to the resulting fine-tuned model as ReAct-Finetune. To run the agent, we prompt it with a goal and initial observation, then alternate assistant messages (for reasoning and action) with user messages (for new observations).

Tab. 6 shows that ReAct-Finetune slightly outperforms the in-context agent on ALFWorld (0.96 vs. 0.93) and Wordcraft (0.74 vs. 0.69), while performing slightly worse on InterCode-SQL (0.79 vs. 0.82). These results suggest that self-collected examples are effective not only for in-context prompting but also for creating competitive fine-tuned agents.

## G Can Databases Transfer Across LLMs?

To evaluate whether our self-improvement approach captures fundamental task structure or merely exploits model-specific artifacts, we conducted additional experiments using Mixtral 8x7B Instruct, a popular open-source model with different architecture and capabilities than GPT-4o-mini. This experiment tests a challenging transfer scenario: databases collected with GPT-4o-mini are used to improve a different model at test time, without any additional data collection or fine-tuning.

### G.1 Experimental Setup

We evaluate Mixtral 8x7B on all three benchmarks, comparing the Fixed-DB baseline to Traj-Bootstrap+DB+Exemplar-Curation. Critically, all databases were previously collected using GPT-4o-mini trajectories. Each configuration is evaluated over 5 random seeds, with results averaged across trials. As seen in Table 7, despite the distribution gap between the collection model (GPT-4o-mini) and deployment model (Mixtral 8x7B), our self-improvement methods yield substantial performance gains across all benchmarks.

| Benchmark | Fixed-DB | TrajBS+DB+Exemplar-Cur | Improvement |
|---|---|---|---|
| ALFWorld | 0.27 | 0.55 | +28 points |
| IC-SQL | 0.52 | 0.70 | +18 points |
| Wordcraft | 0.40 | 0.52 | +12 points |

Table 7: Cross-model generalization results with Mixtral 8x7B Instruct v0.1 using databases collected with GPT-4o-mini. All results averaged over 5 trials. Full Method refers to Traj-Bootstrap + DB-Curation + Exemplar-Curation.

### G.2 Comparison with In-Distribution Performance

| Benchmark | GPT-4o-mini Improvement | Mixtral 8x7B Improvement |
|---|---|---|
| ALFWorld | +20 points | +28 points |
| IC-SQL | +7 points | +18 points |
| Wordcraft | +14 points | +12 points |

Table 8: Comparison of performance improvements (Fixed-DB vs. Traj-BS+DB+Exemplar-Cur) between the collection model (GPT-4o-mini) and a different deployment model (Mixtral 8x7B).

Table 8 compares the improvements observed with Mixtral 8x7B against those achieved with GPT-4o-mini (the collection model) using the same methods. The improvements with Mixtral 8x7B match or exceed those observed with GPT-4o-mini on all three benchmarks. This suggests that our approach captures fundamental task structure—such as effective exploration strategies, common failure modes, and successful action sequences—rather than exploiting model-specific quirks or artifacts.

## H   Can We Predict Agent Success Rates?

We have shown that increasing the number of self-collected examples improves agent performance. Here, we test whether the same examples can also predict performance on new tasks.

On InterCode-SQL and Wordcraft, task difficulty is partly observable from the goal $g$ and initial observation $o_1$. For InterCode-SQL, $g$ is a natural-language query. For Wordcraft, $g$ is the desired element and $o_1$ specifies available crafting elements. In contrast, ALFWorld task difficulty depends heavily on scene layout, which $g$ and $o_1$ do not reveal. We therefore exclude ALFWorld from this analysis.

We use the same embedding model as in retrieval (all-MiniLM-L6-v2 [43]) to encode the concatenated string $[g; o_1]$. We train a calibrated Random Forest classifier to predict task success/failure, calibrating its outputs via 5-fold cross-validation with a learned sigmoid function. For each of 5 independent Traj-Bootstrap trials, we evaluate (1) the classifier's AUROC on held-out tasks, and (2) its calibration.

As shown in Fig. 5, prediction performance improves as the database grows. For InterCode-SQL, AUROC rises from 0.60 (100 tasks) to 0.77 (800 tasks). Wordcraft shows a similar trend, improving from 0.50 (100 tasks) to 0.71 (4000 tasks). In both cases, predictive accuracy increases alongside task performance.

Fig. 6 shows the calibration of the final classifiers (trained on all available training tasks). Predicted success probabilities closely match observed success rates, indicating well-calibrated models.

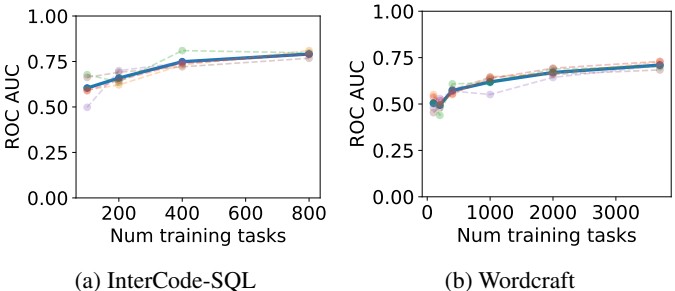

(a) InterCode-SQL     (b) Wordcraft

Figure 5: **AUROC of success prediction improves with more self-collected examples.** Performance continues to rise with increasing database size.

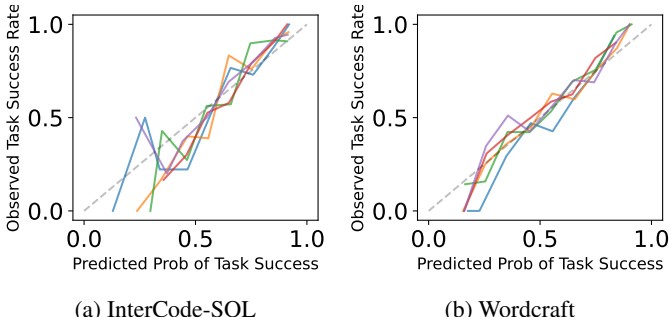

(a) InterCode-SQL     (b) Wordcraft

Figure 6: **Predicted probabilities are well-calibrated.** For both benchmarks, predicted and empirical success rates generally align.

## I   Is it Possible to Bootstrap an Agent Without Initial Hand-Crafted Examples?

Providing a small number of hand-crafted in-context examples is standard practice in the LLM agent literature [9, 13, 12, 4, 10]. However, what if we initialized Traj-Bootstrap with an empty database? In order to understand the value of the initial human-provided examples, we test Traj-Boostrap with and without the initial human-provided examples on Wordcraft. We refer to the variant initialized with an empty database as -Human-Examples. Traj-Bootstrap, initialized by default with a database of 5 human-provided trajectories for Wordcraft, achieves better performance with these starting examples than when initialized from an empty database (-Human-Examples). Performance still scales with database size for -Human-Examples–but in this case fails to reach the performance achieved via 5 human-provided examples, even after self-collecting trajectories on 4000 training tasks. On at least this one task, the initial human-provided trajectories shaped the reasoning and action patterns of the agent in a way that boosted the continual database construction process. We leave exploration of hand-crafting these in-context examples to future work.

## J   Key Details of Prior Agentic Approaches

### J.1   How does Automanual Leverage Hand-Crafted Information

Rather than learning from self-collected examples, an alternate approach to agent construction is to leverage practitioner domain knowledge. Beyond implementing both a hierarchical learning system and code-based action spaces, Automanual [4] incorporates domain knowledge about the ALFWorld task into multiple components of the algorithm. In this section we include some code from the official Automanual GitHub to illustrate.

**Observation spaces**   : Automanual uses a modified observation space that enhances the ALFWorld string by adding two critical pieces of information: 1) The current location of the agent, 2) What the

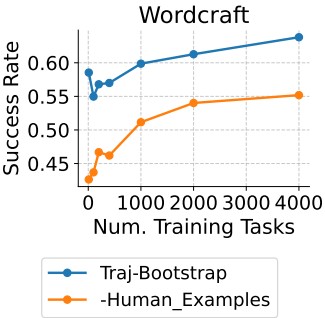

Figure 7: **Ablating the value of initial human-provided examples, Wordcraft.** Traj-Bootstrap, initialized by default with a database of 5 human-provided trajectories for Wordcraft, achieves better performance with these starting examples than when initialized from an empty database (-Human-Examples). Performance still scales with database size for -Human-Examples–but in this case fails to reach the performance achieved via 5 human-provided examples, even after self-collecting trajectories on 4000 training tasks.

agent is currently holding. Both of these pieces of information typically have to be deduced from the trajectory of previous observations and actions, but Automanual tracks them explicitly:

```python
if "Nothing happens" not in observation:
    self.last_obs = observation
    if "go to" in script:
        self.cur_loc = re.search(r'go to (\S+)', script).group(1)
        self.cur_loc_info = observation
    if "open" in script or "close" in script:
        self.cur_loc_info = observation
    if "take" in script:
        self.holding = re.search(r"(?<=take\s)(.*?)(?=\sfrom)",
            script).group(1)
        self.cur_loc_info = ""
    if "put" in script:
        self.holding = "nothing"
        self.cur_loc_info = ""
elif "go to" in script:
    loc = re.search(r'go to (\S+)', script).group(1)
    if loc == self.cur_loc:
        observation = self.cur_loc_info
observation += f" You are at {self.cur_loc} and holding {self.holding}."
```

**Action spaces** : In ALFWorld, any task typically involves three main components: 1) Searching for an object, 2) Performing an action with the object (heating, cooling, cleaning, etc.), 3) Placing the object somewhere. Automanual significantly simplifies both the search and placement operations by providing multi-action helper functions within its code-based action space:

```python
# Define a helper method to find object that is needed
def find_object(agent, recep_to_check, object_name):
    for receptacle in recep_to_check:
        observation = agent.go_to(receptacle)
        # Check if we need to open the receptacle. If we do, open it.
        if 'closed' in observation:
            observation = agent.open(receptacle)
        # Check if the object is in/on the receptacle.
        if object_name in observation:
            object_ids = get_object_with_id(observation, object_name)
            return object_ids, receptacle
    return None, None

```

```
14  # Define a helper method to put object in/on the target receptacle
15  def go_to_put_object(agent, target_receptacle, object_id):
16      observation = agent.go_to(target_receptacle)
17      # check if target_receptacle is closed. If so, open it.
18      if 'closed' in observation:
19          observation = agent.open(target_receptacle)
20      observation = agent.put_in_or_on(object_id, target_receptacle)
21      return observation
```

## J.2 A note on training and test sets

The distinction between how different techniques leverage data is crucial in understanding the generalization capabilities of LLM agents. We can categorize existing approaches based on how they treat training and test data:

**Single-Task Optimization** : Some approaches focus exclusively on improving performance on a single task instance without concern for generalization. For example, Shinn et al. [25] leverages feedback from failed attempts to incrementally improve performance on the same task. Similarly, search methods [22, 24] expand the solution space for a specific problem instance. While these approaches can solve individual tasks, they don't transfer knowledge across different problems, essentially 'overfitting' to a single instance.

**Mixed Train-Test Evaluation** : Some recent work blurs training and test boundaries. For instance, RAP [10] makes multiple passes over the same dataset, allowing the system to 'learn' from some test examples before evaluating on others within the same set. This approach does not assess true generalization capability, as the model has indirect exposure to the test distribution during its learning phase.

**Full Train-Test Separation** : Several papers maintain a clear separation between training and test data: 1) ExpeL [13] extracts general rules from a training set of trajectories and applies them to entirely separate test tasks, 2) AutoGuide [12] generates contextual guidelines from training experiences that are evaluated on distinct test scenarios. 3) AutoManual [4] constructs hierarchical 'manuals' from training interactions that are then applied to novel test tasks.

Our approach similarly ensures that trajectories used for database construction come exclusively from designated training tasks, with evaluation conducted on a separate set of test tasks never seen during the database construction phase. This separation is essential for validating that the knowledge captured by the agent generalizes to new problems rather than memorizing specific solutions.

# K   Comparison to Hand-Crafted InterCode-SQL Agent

We further contextualize the performance of Traj-Bootstrap by comparing to two hand-crafted agents on InterCode-SQL. The Intercode-SQL paper [38] provides a hand-crafted agent, GameSQL to solve the task, and optionally provides the agent with a 'handicap'–giving the agent information on all relevant parts of the database schema. We denote the assisted version as GameSQL+Cheat. Neither agent provides in-context examples, and both share a bespoke, hand-crafted prompt (see App. N).

As seen in Tab. 9, Fixed-DB performs similarly to GameSQL (0.74 vs 0.73), and the performance of our best method, +DB+Exemplar-Curation, approaches the performance of GameSQL+Cheat (0.82 vs 0.84). Therefore, our database-construction techniques lift the performance of a generic ReAct-style agent nearly as much as the lift provided to the hand-crafted agent via 'handicap' access to the database schema.

| Method | Intercode-SQL Success Rate |
|---|---|
| GameSQL | 0.73 |
| GameSQL+Cheat | **0.84** |
| Fixed-DB | 0.74 |
| Traj-Bootstrap | 0.79 |
| +DB-Curation | 0.78 |
| +Exemplar-Curation | 0.81 |
| +DB+Exemplar-Curation | **0.82** |

Table 9: **Comparison of agent success rates on InterCode-SQL: contextualizing the performance of Traj-Boostrap.** Without cheats, the hand-crafted GameSQL agent (0.73) performs comparably to Fixed-DB (0.74). With handicap access to the database schema, GameSQL+Cheat (0.84) slightly outperforms +DB+Exemplar-Curation (0.82). The boost from our database-construction techniques nearly matches the boost from providing the GameSQL agent with access to privileged database schema information.

# L    Benchmark Details

## L.1    ALFWorld

ALFWorld [37] is a text-based environment that aligns with embodied tasks, allowing agents to navigate and manipulate objects through textual commands. We use the standard ALFWorld benchmark consisting of 3500 training tasks and 134 out-of-distribution test tasks across 6 task categories:

- Pick & Place: Find and move objects to specified locations
- Clean & Place: Find, clean, and place objects
- Heat & Place: Find, heat, and place objects
- Cool & Place: Find, cool, and place objects
- Pick Two & Place: Find and move two objects to a specified location
- Look at Object: Find an object and examine it under light

Following [10], for the ALFWorld benchmark we perform similarity search over task categories in addition to the other retrieval keys (goal, plan, observation, action). We do this to follow the convention in this prior work.

For our initial human-provided exemplars, we used the 18 successful trajectories (3 per task category) provided by Zhao et al. [13]. These trajectories were used to initialize all database instances.

The success criteria for ALFWorld tasks are defined by the environment and require the agent to satisfy all conditions specified in the goal. For example, in a 'Heat & Place' task, the agent must find the target object, place it in the microwave, turn on the microwave, and finally place the heated object at the specified destination. Both Autoguide and Automanual allow 50 actions for task completion–but choosing to employ "reasoning" counts as an action. Since we force our agent to reason at every step, we allow our agents (Fixed-DB, Traj-Bootstrap and variants) only 30 steps for task completion (on Autoguide and Automanual, the agent does not reason in practice at most steps ex. in a search procedure).

For this benchmark, we do not provide an action space string to the LLM, relying purely on the in-context examples to communicate the action space.

## L.2    InterCode-SQL

InterCode-SQL [38] is an interactive coding environment for evaluating language agents' SQL programming abilities. We use a subset of the InterCode benchmark focusing on SQL query generation, built upon the Spider SQL dataset. Of the 1034 tasks in the dataset, we randomly assign 800 tasks to train and the remaining 234 tasks to test.

Each task provides a natural language query request. The agent must generate a syntactically correct SQL query that retrieves the requested information. The agent must first execute queries to understand the database schema. The environment provides feedback on syntax errors and execution results, but the agent is only allowed to submit a solution once.

The success criteria for InterCode-SQL tasks require the agent to submit a solution query within 10 steps. The environment executes the query and compares the results against a ground-truth reference.

For our initial human-provided exemplars, we collected 10 human-created trajectories for 10 randomly-selected training tasks. These trajectories were used to initialize all database instances. For all solved trajectories, we append the solution query to the goal string–since the goal of the task is to 'discover' this query through interacting with the SQL database.

We used the following action space string for InterCode-SQL:

```
Your action space is outputting valid mysql commands to solve the sql task.
You will be evaluated on the Latest Standard Output.
If you believe the latest observation is the final answer, you can complete the
    task by running 'submit' by itself.
You have 10 iterations to solve the task.
Follow the syntax and logical flow from the provided examples exactly.
```

## L.3 Wordcraft

Wordcraft [39] is a simplified adaptation of the game Little Alchemy, where agents must combine elements to create new elements through multi-step processes. We randomly select 4000 training tasks and 500 test tasks from the subset of tasks requiring up to 2 steps to solve, with the train-test split separating the tasks into disjoint sets of goal elements.

The agent starts with a set of elements and must discover combinations that creates a particular target element specified in the goal. The environment provides feedback on successful combinations and updates the available elements accordingly.

The success criteria for Wordcraft tasks require the agent to create the target element within 4 steps, while the minimum solution length is up to 2 steps.

For our initial human-provided exemplars, we collected 4 human-annotated trajectories from randomly-selecting training tasks. These trajectories were used to initialize all database instances. We collected fewer initial trajectories for Wordcraft than for InterCode-SQL (4 vs 10) since Wordcraft is a slightly simpler task, requiring up to 4 steps for task completion while InterCode-SQL requires up to 10.

We used the following action space string for Wordcraft:

```
Output strings with the names of the two entities we would like to combine in this
    step.
```

## L.4 Note on Benchmark Selection

We selected three sequential decision-making benchmarks that cover different reasoning challenges—**ALFWorld** [37] tests text-based navigation and object manipulation, **InterCode-SQL** [38] tests interactive code generation, and **Wordcraft** [39] tests compositional reasoning.

While prior works [13, 12] test on WebShop [45], we encountered bugs in generating achievable goals on the full benchmark (confirmed by https://github.com/princeton-nlp/WebShop/issues/43) and identified tasks with incorrect rewards.

We excluded QA benchmarks (HotPotQA [46], etc.) because performance depends on information retriever quality and LLM self-evaluation efficacy, two factors that would confound our study of LLM Self-Improvement. We plan to test our algorithms on QA benchmarks in future work.

## M Computational Resources

All experiments were conducted using the following computational resources:

- 1 NVIDIA A5000 GPU (24GB memory) for embedding computation
- 64GB RAM

The majority of computation was spent on OpenAI API calls for the LLM-based decision-making. Database operations including embedding computation, storage, and retrieval accounted for less than 5% of the total computation time.

For embedding computations, we used all-MiniLM-L6-v2 [43].

For LLM inference, we used the OpenAI API for GPT-4o-mini, which required approximately:

- 2,000,000 API calls for ALFWorld
- 200,000 API calls for InterCode-SQL
- 500,000 API calls for Wordcraft

The total cost of API usage was approximately $3,000 USD.

## N GameSQL Prompt

Yang et al. [38] write this hand-crafted prompt for the GameSQL agent:

```
1  '''
2  {self.language}Env` is a multi-turn game that tests your ability to write
3  a {self.language} command that produces an output corresponding to a natural
   ↪  language query.
4
5  ## GAME DESCRIPTION
6  At the start of this game, you are given a natural language query describing
   ↪  some
7  desired output (i.e. "Find the first name of a student who have both cat and
   ↪  dog pets").
8  Aside from the natural language query, you have no information about the
   ↪  tables you have access to.
9
10 The game will be played in a series of turns. Each turn, you can submit a
   ↪  {self.language} command.
11 You will then get a response detailing the output of your {self.language}
   ↪  query along with a reward
12 that tells you how close your {self.language} command is to the correct
   ↪  answer.
13
14 The goal of this game is to write a {self.language} command that gets a reward
   ↪  of 1. The game will automatically
15 terminate once you get a reward of 1.
16
17 ## INPUT DESCRIPTION
18 Each turn, you can submit a {self.language} command. Your {self.language}
   ↪  command should be formatted as follows:
19
20 ```{self.language}
21 Your {self.language} code here
22 ```
23
24 Your {self.language} command can help you do one of two things:
25 1. Learn more about the tables you have access to
26 2. Execute {self.language} commands based on these tables to generate the
   ↪  correct output.
27
```

```
28   ## OUTPUT DESCRIPTION
29   Given your {self.language} command input, `{self.language}Env` will then give
     ↪   back output formatted as follows:
30
31   Output: <string>
32   Reward: <decimal value between 0 and 1>
33
34   The output is a string displaying the result from executing your
     ↪   {self.language} query.
35   The reward is a decimal value between 0 and 1.
36
37   ## REWARD DESCRIPTION
38   The reward should be interpreted as a ratio. It tells you how many rows your
     ↪   {self.language}
39   command outputted correctly compared to the correct answer.
40
41   ## RULES
42   1. Do NOT ask questions. Your commands are fed directly into a SQL compiler.
43
44   ## STRATEGY
45   You are free to play as many turns of the game as you'd like to inspect tables
46   and develop your {self.language} command.
47
48   The best strategy for this game is to first write {self.language} commands
     ↪   that help you learn
49   about the tables that you have access to. For instance, in a SQL environment,
     ↪   you might use `SHOW TABLES`
50   and `DESC <table name>` to learn more about the tables you have access to.
51
52   Once you have a good understanding of the tables, you should then write
     ↪   {self.language} commands
53   that would answer the natural language query using the tables you have access
     ↪   to.
54   '''
```

