# OpenReview forum: "Self-Generated In-Context Examples Improve LLM Agents for Sequential Decision-Making Tasks"
_NeurIPS.cc/2025/Conference — NeurIPS 2025 poster_

### Official Review · Reviewer_YEwY · 2025-07-02

**Clarity:** 3
**Significance:** 3
**Originality:** 3
**Rating:** 5
**Confidence:** 4

**Summary:**

This paper proposes an exemplar curation approach for LLM agents in sequential decision making tasks. The proposed approach views the training of LLM agents as exemplar curation, where a growing list of exemplars are curated using two complementary methods, DB-curation,  and exemplar-curation, to maintain a database of retrievable successful examples that the agent has attempted. The proposed approach is evaluated on three sequential decisionmaking tasks, showing improvements competitive with the state of the arts. Ablation studies validate the effectiveness of proposed curation approaches.

**Questions:**

- Does the agent see few-shot exemplars aside from the ones it retrieves?
- Can you provide examples of retrieved exemplars? It would be useful to show how the retrieved exemplars collective shape the agent's policy.
- Is the high level plan fixed once generated? How important is it for the initial plan to be correct?
- Is any tradeoff for efficiency (e.g. not retrieving at every step, retrieving only once every k steps) possible?

**Ethical Concerns:**

["NO or VERY MINOR ethics concerns only"]

**Final Justification:**

The paper proposes a simple but effective method for learning as trajectory accumulation, which takes a different approach to many other agents in this domain, which focus on complex agent scaffolds with hand-crafted exemplars. It is likely that approaches such as the proposed method will likely scale better with increasingly more capable models, which makes the paper's contribution meaningful.

My main concern was that the method requires excessive computational costs or unrealistic assumptions about train-test distribution overlap, which the authors have addressed successfully. Assuming that the discussions from rebuttal are fully reflected in the updated draft, I am happy to give it an accept rating.

**Limitations:**

yes

**Quality:**

3

**Strengths And Weaknesses:**

Strengths
- The proposed approach shows strong performance across all evaluated tasks.
- The idea of training as growing a database of retrievable knowledge is interesting, and salient for LLM agents, which are particularly effective in retrieval-augmented usage, and cannot easily be finetuned. This is also related to the idea of episodic memory in cognitive science, which is an interesting connection.
- The components of the proposed approach, including the curation approaches and multi key retrieval method, are well designed.
- The method appears to make only minor task-specific modifications to ReAct, making it not overly complicated, and generally applicable.

Weaknesses
- The method appears computationally expensive, especially when maintaining N(x2, when using both curation approaches together) databases, and training over hundreds or thousands of examples. Computational costs (including the training as well as retrieval costs at every step), are not analyzed thoroughly, so we cannot be certain that the costs are justified.
- From the performance results, it appears that significant gains come from accruing a potentially critical mass of training exemplars, enough to cover the test distribution. For example, while ALFWorld is complex, its test set consists of 134 examples across 6 task types. I would not be surprised if the retrieved exemplars are extremely similar to the actual test task. That said, I think it would still be an interesting finding if this were the case.
- The results are limited to GPT4o-mini, limiting generalizability.

---

> ### Author Rebuttal · Authors · 2025-07-31
>
> Thank you for your thoughtful review and positive assessment. We are excited that you liked our approach's strong performance, the connection to episodic memory in cognitive science, and appreciated how the design of our components work together to maintain general applicability with minimal task-specific modifications.
>
> > Computational costs (including the training as well as retrieval costs at every step), are not analyzed thoroughly, so we cannot be certain that the costs are justified.
>
> **Cost Analysis:** We provide detailed token usage to address your cost-benefit question. Average tokens per request: 5,047/68 input/output tokens (Wordcraft), 5,385/37 (InterCode), 3,706/30 (ALFWorld), with a maximum episode of 8, 20, and 61 requests respectively. More generally, for any task, we require 2 requests per step (reasoning + acting), plus 1 additional request when using planning (see Appendix D6, L550-556). At GPT-4o-mini pricing (0.15/0.6 USD per 1M tokens), cost per episode ranges from 0.004 USD for InterCode to 0.035 USD for ALFWorld. To collect 1000 episodes, maximum API costs range from 4-35 USD.
>
> **Contextualizing the costs of scaling performance along various axes:** In Section 6.3, we demonstrate that our approach can yield substantial cost savings to two common alternatives for scaling model performance: (i) switching to a larger LLM, and (ii) test-time scaling via multiple attempts per task (assuming access to a perfect verifier). On ALFWorld, our full configuration of Traj-Bootstrap + DB-Curation + Exemplar-Curation with 5 parallel databases requires 3,500 × 5 training trajectories. Using GPT-4o-mini, the worst-case cost of constructing the databases is 600 USD, after which per-task `test-time' cost of using GPT-4o-mini is only ~0.034 USD. By comparison, using GPT-4o at test time (without collecting training trajectories) costs ~0.57 USD per task, reaching the same cumulative spend as our approach after ~**1,100** test-time tasks. The alternative baseline of test-time scaling with 3 attempts per task costs a total of ~0.10 USD per task, which exceeds our method's total cost after ~**8,900** test-time tasks. In other words, after a modest up-front investment (of a few hundred dollars of compute) our approach remains **consistently more cost-effective** than either alternative, while also improving test-time agent performance. Consider a hypothetical production agentic service processing a million ALFWorld tasks a day (a very modest industry-scale production throughput). The cost savings **in a single day** of using our method, including all offline dataset construction, would be 530,000 USD vs. GPT-4o, or 60,000 USD vs GPT-4o-mini with 3 trials of test time compute.
>
> > Is any tradeoff for efficiency (e.g. not retrieving at every step, retrieving only once every k steps) possible?
>
> Retrieving every 5 steps instead of every step on ALFWorld leads to a drop in task success rate when comparing Fixed-DB baseline (0.69 vs 0.73 with every-step retrieval) and our best method Traj-Bootstrap+DB-Curation+Exemplar-Curation (0.91 vs 0.93 with every-step retrieval). This represents only 4-point and 2-point decreases respectively—a performance trade-off for substantial efficiency gains. This 5× reduction in retrieval frequency only reduces retrieval operations (from 32 to 7 per episode for ALFWorld's worst-case scenario). While performing retrieval less frequently doesn't reduce LLM token usage per call, it may significantly reduce LLM inference cost in systems with prefix caching (due to the shared prompt structure for 5 consecutive LLM invocations)
>
> > The results are limited to GPT4o-mini, limiting generalizability.
>
> We conducted additional experiments using Mixtral 8x7B Instruct v0.1. Using databases collected with GPT-4o-mini, we compared Fixed-DB baseline versus our best method when evaluated with Mixtral. Results averaged across 5 trials show strong cross-model generalization: ALFWorld (0.55 vs 0.27, +28 points), IC-SQL (0.70 vs 0.52, +18 points), Wordcraft (0.52 vs 0.40, +12 points). These improvements match or exceed our GPT-4o-mini results, demonstrating that curated databases transfer effectively across model families and addressing generalizability concerns.
>
> > From the performance results, it appears that significant gains come from accruing a potentially critical mass of training exemplars, enough to cover the test distribution. For example, while ALFWorld is complex, its test set consists of 134 examples across 6 task types. I would not be surprised if the retrieved exemplars are extremely similar to the actual test task. That said, I think it would still be an interesting finding if this were the case.
>
> You are correct that significant gains come from accruing a progressively larger, diverse set of training exemplars, and the algorithmic mechanism by which the Traj-Bootstrap algorithm improves performance with scale is that it becomes more likely for a training exemplar to have similarities to a later test-time task. However, note that on the benchmarks tested, there is a train-test split between the tasks. For instance, ALFWorld is evaluated on the eval-out-of-distribution split, **an entirely different set of room/object layouts than the layouts present in the training set (L659)**. Therefore, the LLM cannot simply copy the actions from the train tasks to the test tasks.
>
> > Does the agent see few-shot exemplars aside from the ones it retrieves?
>
> The agent is only provided retrieved trajectories—no additional few-shot examples beyond what's retrieved from our curated database. Full prompt templates are in Appendix D.2.
>
> > Can you provide examples of retrieved exemplars? It would be useful to show how the retrieved exemplars collective shape the agent's policy.
>
> We will include concrete examples of retrieved trajectories in our Appendix to demonstrate how they collectively shape agent behavior across different scenarios. Unfortunately, we cannot include screenshots or other media in the rebuttal response this year, and showing a significant sample of trajectories in this response is not feasible due to character limits, our trajectories are similar to the ones seen as in-context examples in the ExpeL paper [1].
>
> > Is the high level plan fixed once generated? How important is it for the initial plan to be correct?
>
> In our ReAct-style implementation, the high-level plan is generated once per task—but the agent reasons step-by-step with retrieved trajectories providing guidance for both immediate actions and longer-term reasoning patterns. The plan is important to guiding the trajectory on the longer-horizon ALFWorld task, but is omitted altogether for the shorter horizon InterCode-SQL and Wordcraft tasks. Please see Appendix D6, L550-556 for details.
>
> We will incorporate these clarifications and additional results in our revision.
>
> [1] Zhao, Andrew, et al. "Expel: Llm agents are experiential learners." Proceedings of the AAAI Conference on Artificial Intelligence. Vol. 38. No. 17. 2024.

---

> > ### Comment · Reviewer_YEwY · 2025-08-05
> >
> > I appreciate the detailed response from the authors, and they have addressed my main concerns. I have raised my score accordingly. The authors should include these analyses in the updated draft, as it would make the paper much stronger by showing that the upfront costs to construct a trajectory DB is worthwhile.

---

> > > ### Author Response · Authors · 2025-08-05
> > >
> > > Thank you for your feedback! We will definitely include these analyses in the updated draft.

---

### Official Review · Reviewer_HntP · 2025-07-03

**Clarity:** 2
**Significance:** 2
**Originality:** 2
**Rating:** 4
**Confidence:** 3

**Summary:**

The paper proposes an approach to improve LLM Agent by retrieving and curating existing trajectories. The proposed method is based on first collecting successful agent trajectories, then improve the data quality by database-level curation or exemplar-level curation. Experiment shows that each proposed component is effective on ALFWorld, InterCode-SQL and Wordcraft in terms of improving the success rate of LLM after seeing more training trajectories.

**Questions:**

- Is it necessary to distinguish database-level and exemplar-level curation? Is it okay to simplify the organization and just do atomic-level data curation with a unified framework?
- Table 2 uses different LLM models for baselines and proposed method, it would be nice to make them consistent.
- Table 2 show the bootstrap method's performance with gpt-4o-mini. Can the authors include some results from other models (e.g. open-sourced models) to justify the generalizibility of the method?
- Is the retrieval on trajectory-level or sub-trajectory-level?

**Ethical Concerns:**

["NO or VERY MINOR ethics concerns only"]

**Final Justification:**

The authors included more experiment results show the generalizability of the method. In the case where the agent is supposed to interact with a large amount of test tasks, the proposed method can be cost-efficient.

I still think that the presentation of the paper can be improved. For example, show some concrete examples of database and exemplar. Also, it's better to incorporate the discussion in rebuttal into the final paper.

**Limitations:**

Limitations are discussed in Appendix A, about the method may be less effective with weaker models.

**Quality:**

2

**Strengths And Weaknesses:**

## Strengths
- The experiments include full ablation study of each single component in algorithm, making it easy to interpret the method and results.
- The paper writing is clear and the result presentation is good.

## Weaknesses
- The novelty of the proposed method seems to be limited. The main idea of this paper is similar to a COLM paper [Xiao 2024], which also improves LLM agents by distilling knowledge from past trajectories, and also does automatic task decomposition + learning from failed trajectories. From the results on ALFWorld, these two methods are on par, but [Xiao 2024] uses fewer training trajectories in total. It would be nice if the authors can justify the novelty, uniqueness and advantages of the proposed method, and do some experiments to compare with them.
- Regarding test-time costs, it would be nice if the authors can also provide details of total number of input tokens and additional computations required for retrieval, in comparison to baseline methods.



Xiao et al. O3D: Offline Data-driven Discovery and Distillation for Sequential Decision-Making with Large Language Models. COLM 2024.

---

> ### Author Rebuttal · Authors · 2025-07-31
>
> Thank you for your detailed review and feedback. We appreciate that you thought our experiments include a "full ablation study of each single component" and that the "paper writing is clear and the result presentation is good."
>
> > The main idea of this paper is similar to a COLM paper [Xiao 2024], which also improves LLM agents by distilling knowledge from past trajectories, and also does automatic task decomposition + learning from failed trajectories
>
> We appreciate you highlighting the relationship with O3D, as it helps clarify our distinct contributions. While both works focus on improving LLM agents through past trajectories, our approaches are fundamentally different in philosophy and mechanism.
>
> **Key Algorithmic Differences:** O3D employs data distillation, summarization, task decomposition, and explicit learning from failed trajectories. In contrast, our method intentionally avoids these complex mechanisms, instead focusing on collecting, filtering, and retrieving raw low-level trajectories. Our core hypothesis is that scaling the number of trajectories available to retrieve in-context and applying thoughtful data curation achieves performance superior to state-of-the-art methods requiring complex hierarchical learning and extensive manual prompt engineering—and our results support this claim.
>
> **Performance and Model Efficiency:** Our approach achieves superior performance (93% with GPT-4o-mini for ours vs 91% with GPT-4 for O3D on ALFWorld). This demonstrates a key advantage: we can lift weaker models to strong performance purely through database scaling, without human engineering, manual prompting, or fine-tuning. Of course, improved performance with a weaker base model requires a tradeoff somewhere–and in this case that tradeoff is using more LLM calls at training time–not test time–to create a larger database of (self-collected) trajectories.
>
> In our analysis of computational costs (included later in this rebuttal), a hypothetical production agentic service processing a million ALFWorld tasks a day (a very modest industry-scale production throughput) could save in a single day of using our method, including all offline dataset construction, 530,000 USD vs. GPT-4o, or 60,000 USD vs GPT-4o-mini with 3 trials of test time compute.
>
> **Methodological Contributions:** Our work makes two distinct methodological contributions: (1) demonstrating how scaling the number of trajectories available to retrieve in-context and applying thoughtful data curation can exceed state-of-the-art methods that utilize complex hierarchical learning and extensive manual prompt engineering, and (2) presenting a novel retrieval scheme--with multi-key retrieval, re-retrieval at each step of a trajectory, leveraging similarity across “reasoning” rather than “observation”, see Appendix C for details-- extracts significant value from simple data. This challenges the assumption that complex data processing is necessary for trajectory-based improvement.
>
> > Can the authors include some results from other models (e.g. open-sourced models) to justify the generalizibility of the method?
>
> Addressing your request for results from other models, we conducted additional experiments using Mixtral 8x7B Instruct v0.1, a popular open-source model. Using databases previously collected with GPT-4o-mini, we compared performance when Mixtral 8x7B uses our Fixed-DB baseline versus our best method (Traj-Bootstrap + DB-Curation + Exemplar-Curation). Results averaged across 5 trials demonstrate strong improvements: ALFWorld shows a 28-point boost (0.55 vs 0.27), IC-SQL achieves an 18-point gain (0.70 vs 0.52), and Wordcraft exhibits a 12-point improvement (0.52 vs 0.40). These improvements match or exceed our GPT-4o-mini results, demonstrating cross-model generalization and broader applicability.
>
> > Regarding test-time costs, it would be nice if the authors can also provide details of total number of input tokens and additional computations required for retrieval, in comparison to baseline methods.
>
> **Cost Analysis:** We provide detailed token usage to address your cost-benefit question. Average tokens per request: 5,047/68 input/output tokens (Wordcraft), 5,385/37 (InterCode), 3,706/30 (ALFWorld), with a maximum episode of 8, 20, and 61 requests respectively. More generally, for any task, we require 2 requests per step (reasoning + acting), plus 1 additional request when using planning (see Appendix D6, L550-556). At GPT-4o-mini pricing (0.15/0.6 USD per 1M tokens), cost per episode ranges from 0.004 USD for InterCode to 0.035 USD for ALFWorld. To collect 1000 episodes, maximum API costs range from 4-35 USD.
>
> **Contextualizing the costs of scaling performance along various axes:** In Section 6.3, we demonstrate that our approach can yield substantial cost savings to two common alternatives for scaling model performance: (i) switching to a larger LLM, and (ii) test-time scaling via multiple attempts per task (assuming access to a perfect verifier). On ALFWorld, our full configuration of Traj-Bootstrap + DB-Curation + Exemplar-Curation with 5 parallel databases requires 3,500 × 5 training trajectories. Using GPT-4o-mini, the worst-case cost of constructing the databases is 600 USD, after which per-task `test-time' cost of using GPT-4o-mini is only ~0.034 USD. By comparison, using GPT-4o at test time (without collecting training trajectories) costs ~0.57 USD per task, reaching the same cumulative spend as our approach after ~**1,100** test-time tasks. The alternative baseline of test-time scaling with 3 attempts per task costs a total of ~0.10 USD per task, which exceeds our method's total cost after ~**8,900** test-time tasks. In other words, after a modest up-front investment (of a few hundred dollars of compute) our approach remains **consistently more cost-effective** than either alternative, while also improving test-time agent performance. Consider a hypothetical production agentic service processing a million ALFWorld tasks a day (a very modest industry-scale production throughput). The cost savings **in a single day** of using our method, including all offline dataset construction, would be 530,000 USD vs. GPT-4o, or 60,000 USD vs GPT-4o-mini with 3 trials of test time compute.
>
> > Is it necessary to distinguish database-level and exemplar-level curation? Is it okay to simplify the organization and just do atomic-level data curation with a unified framework?
>
> **Regarding database vs. exemplar-level curation:** These serve complementary purposes—database-level curation propagates the best populations of training examples while exemplar-level curation identifies high-utility individual examples. Our ablations show both contribute meaningfully to performance.
>
> > Is the retrieval on trajectory-level or sub-trajectory-level?
>
> When planning, retrieval occurs at the trajectory level (L486-488). At each step, we retrieve a sub-trajectory window around the most relevant stat3 (L488-490). For a detailed overview of retrieval, see Algorithm 4 in Appendix C.
>
> > Table 2 uses different LLM models for baselines and proposed method, it would be nice to make them consistent.
>
> We understand your concern about model consistency in Table 2. **We use the same LLM models for baselines and proposed methods when possible.** For Automanual, since source code is available, we evaluated two configurations for thoroughness: (1) GPT-4o-mini for both manual generation and inference (apples-to-apples comparison with our method), and (2) GPT-4-turbo for manual generation with GPT-4o-mini for test-set inference (representing their best-case scenario from their original paper while maintaining fair inference comparison). For AutoGuide, we report results from their paper (GPT-3.5-turbo + GPT-4-turbo) since code was not available.

---

> > ### Comment · Reviewer_HntP · 2025-08-05
> >
> > Thank the authors for the detailed response. The additional analysis and experiments are convincing and have addressed my major concerns. I will raise my score accordingly.

---

### Official Review · Reviewer_4WEi · 2025-07-03

**Clarity:** 2
**Significance:** 3
**Originality:** 2
**Rating:** 4
**Confidence:** 4

**Summary:**

This paper proposes a method for improving large language model (LLM) agents on sequential decision-making tasks by leveraging self-generated, successful trajectories as in-context examples. Instead of relying on human-engineered prompts or domain-specific action/observation design, the authors introduce a trajectory bootstrapping approach that enables agents to iteratively build and refine a database of successful trajectories through autonomous experience. Key innovations include:
	•	Traj-Bootstrap: a simple accumulation of successful task completions.
	•	+DB-Curation: database-level curation using population-based training.
	•	+Exemplar-Curation: exemplar-level selection of trajectories using a value metric based on empirical utility.
Extensive experiments on ALFWorld, Wordcraft, and InterCode-SQL demonstrate notable performance gains, even surpassing methods that use more powerful LLMs or handcrafted components.

**Questions:**

1	Paper Structure Revision: The placement of Related Work after Problem Statement is unconventional and makes the reading flow less logical. Consider reorganizing the sections to follow the standard order: Introduction → Related Work → Method → Experiments.
2	Generalization to New Tasks: How does the system handle domain shifts or novel goals that were not part of the training task distribution? The reliance on accumulated in-context trajectories might hurt generalization.
3	Training/Test Gap Concerns: While the paper aims to reduce hand-crafted engineering, the method still depends on hand-curated starting examples and extensive self-training—how is the potential gap between training and deployment (e.g., in online settings) mitigated?

**Ethical Concerns:**

["NO or VERY MINOR ethics concerns only"]

**Final Justification:**

My concerns have been addressed and I am recommending acceptance.

**Limitations:**

Yes

**Quality:**

3

**Strengths And Weaknesses:**

Strengths: The paper offers a practical and scalable alternative to manual prompt engineering, validated through solid empirical results. The bootstrapping and curation strategies are clearly described and effectively evaluated across diverse tasks. The proposed method offers a general, task-agnostic mechanism that could be widely useful in agentic settings. It shows that high performance can be achieved even with lightweight models through data-centric optimization.


Weaknesses: The methodology is well-implemented but somewhat straightforward, relying heavily on repeated LLM calls during training. While empirical gains are strong, theoretical insights are limited. The structure of the paper is confusing. Notably, placing the Problem Statement before Related Work breaks conventional scientific writing flow. This likely reflects an incomplete or unpolished draft and disrupts the narrative logic for readers. The novelty lies in the reuse and filtering of examples, which is more incremental than transformative compared to prior ReAct-style agents and retrieval-based pipelines. The overall conceptual framework builds heavily on existing ideas (ReAct, retrieval-based LLMs, in-context learning), and the innovation is mainly in implementation rather than theory or modeling.

---

> ### Author Rebuttal · Authors · 2025-07-31
>
> Thank you for your detailed review and constructive feedback. We are encouraged you think our method offers "a practical and scalable alternative to manual prompt engineering" with "solid empirical results" and represents "a general, task-agnostic mechanism that could be widely useful in agentic settings."
>
> > 1 Paper Structure Revision: The placement of Related Work after Problem Statement is unconventional and makes the reading flow less logical. Consider reorganizing the sections to follow the standard order: Introduction → Related Work → Method → Experiments.
>
> We respectfully note that Problem Statement sections before Related Work are commonly used in machine learning papers, particularly when the technical problem formulation helps contextualize subsequent related work discussions. In this paper we made the deliberate choice to help readers understand the specific setup before contextualizing our proposed solution with respect to prior work. However, we appreciate your feedback on narrative flow and will ensure the presentation is as clear as possible in our revision.
>
> > 2 Generalization to New Tasks: How does the system handle domain shifts or novel goals that were not part of the training task distribution? The reliance on accumulated in-context trajectories might hurt generalization.
>
> Your question about handling domain shifts and novel goals is insightful. To address generalizability concerns, we conducted additional experiments using Mixtral 8x7B Instruct v0.1, a popular open-source model with different architecture and capabilities than GPT-4o-mini. Using databases previously collected with GPT-4o-mini, we compared performance when Mixtral 8x7B uses our Fixed-DB baseline versus our best method (Traj-Bootstrap + DB-Curation + Exemplar-Curation).
>
> Results averaged across 5 trials demonstrate strong cross-model generalization: ALFWorld shows a 28-point improvement (0.55 vs 0.27), IC-SQL achieves an 18-point gain (0.70 vs 0.52), and Wordcraft exhibits a 12-point boost (0.52 vs 0.40). **These improvements match or exceed the gains we observed with GPT-4o-mini (20, 7, and 14 points respectively), suggesting our approach captures fundamental task structure rather than model-specific artifacts.** This cross-model transfer demonstrates robustness to significant distribution gaps between collection and deployment models.
>
> Although our experimental testbeds do not feature domain shift, handling situations involving domain shift is a strength of our approach by design. Unlike approaches that use a fixed set of in-context examples or approaches that require model fine-tuning (typically a high-cost, high-latency operation), our system allows for continual learning via accumulating examples in the database during deployment, naturally expanding coverage to new domains or task variations encountered in practice. **This "learning while doing" capability means initial domain gaps can be progressively closed, potentially in seconds, by accessing recently collected experience.**
>
> > 3 Training/Test Gap Concerns: While the paper aims to reduce hand-crafted engineering, the method still depends on hand-curated starting examples and extensive self-training—how is the potential gap between training and deployment (e.g., in online settings) mitigated?
>
> You raise an important point about the training-deployment gap. The cross-model experiments above address this concern, as GPT-4o-mini (collection) and Mixtral 8x7B (evaluation) represent a substantial train-test distribution gap in terms of model architecture and capabilities. The strong performance transfer provides evidence that our curated databases generalize effectively across different deployment contexts.
>
> Additionally, while a human-provided starting database is helpful for jump starting performance, it is not strictly necessary—as shown in Appendix G, Figure 7, **our method can actually bootstrap from an empty starting database**, albeit with less data efficiency. This reduces dependence on hand-crafted engineering compared to traditional approaches.
>
> **The key insight is that our system becomes more robust over time. As it encounters new scenarios during deployment, it automatically expands its database with successful solutions, making it increasingly suitable for diverse real-world applications.**
>
> > The overall conceptual framework builds heavily on existing ideas (ReAct, retrieval-based LLMs, in-context learning), and the innovation is mainly in implementation rather than theory or modeling.
>
> We agree completely that our approach builds on existing agentic designs (ReAct, retrieval, in-context learning). We emphasize that our contribution lies in demonstrating that simple, well-designed exemplar curation and retrieval can outperform approaches requiring hand-crafted task-specific observation/action spaces and hierarchical learning components, while being significantly easier to implement and scale. Our core finding—that scaling the number of trajectories available to retrieve in-context and applying thoughtful data curation achieves performance superior to state-of-the-art methods that utilize complex hierarchical learning and extensive manual prompt engineering—represents a valuable insight for the field, offering a practical data-driven path to scaling in-context LLM agent performance without architectural complexity.

---

> > ### Author Response · Authors · 2025-08-05
> >
> > Thank you again for your feedback! As we are close to the end of the discussion window, could you please let us know if there is anything else we could help clarify about our submission?

---

> > > ### Comment · Reviewer_4WEi · 2025-08-06
> > >
> > > Thanks for the rebuttal and my concerns have been addressed.

---

### Official Review · Reviewer_eKr8 · 2025-07-03

**Clarity:** 4
**Significance:** 3
**Originality:** 2
**Rating:** 5
**Confidence:** 4

**Summary:**

This paper provides extensive experiments consistently showing that using only successful self-generated trajectories as in-context examples can cause an increase in performance. Furthermore, they provide a dataset curation method (DB+) to store the most relevant successful trajectories that can be used with multiple reasoning algorithms.

**Questions:**

See Weakness 1

**Ethical Concerns:**

["NO or VERY MINOR ethics concerns only"]

**Final Justification:**

I think the authors did a relatively good job of addressing concerns of cost analysis and failed trajectory use. They also responded well to novelty issues brought up by another reviewer. I think this paper can contribute to the field.

**Limitations:**

See Weakness 1

**Quality:**

3

**Strengths And Weaknesses:**

**Strengths**

The authors use a trajectory bootstrapping method, DB+, that samples successful self-generated trajectories to be used for self-improvement for sequential decision making

Lots of ablations to show consistent increased performance with successful self-generated examples across different algorithms, tasks, and models.

The idea of predicting agent success in Section 6.4 is novel

Line 155 about collective properties is I think a significant benefit of this approach as it relates to the coverage of situations that the agent has examples to rely on

**Weakness**

My main problem is that the paper discards the failed trajectories and leaves how to use them as future work. I understand their message is that performance can improve with just successful trajectories. However, I am worried about the inefficiency of LLM calls when discarding the failed ones. Basically my question is if the improvement is worth the amount of LLM calls? Table 2 shows a 5% increase from 100->3500 training tasks. I understand Figure 2 shows that in some cases only 1000 training tasks are needed to reach the performance of 3500 training tasks. Furthermore, in Figures 2 and 3, of these training tasks, how many of them have failed trajectories that are discarded?

Consider also citing some In-Context RL papers such as [1], which also looks at multi-step decision making.

[1] Dai, Zhenwen, Federico Tomasi, and Sina Ghiassian. "In-context Exploration-Exploitation for Reinforcement Learning." arXiv preprint arXiv:2403.06826 (2024).

---

> ### Author Rebuttal · Authors · 2025-07-31
>
> Thank you for your thorough review and positive assessment of our work. In particular, we were excited that you agree with our perspective on database quality—that "database quality emerges from collective properties—like coverage, diversity, and complementarity across examples—not just individual trajectory quality." (Line 155). We also agree that this is one of the core contributions of our submission.
>
> > My main problem is that the paper discards the failed trajectories and leaves how to use them as future work. I understand their message is that performance can improve with just successful trajectories. However, I am worried about the inefficiency of LLM calls when discarding the failed ones.
>
> We appreciate your concern about discarding failed trajectories and provide detailed analysis below.
>
> **Failed trajectories are used to improve the system:** During database construction, our method achieves success rates of 81.3% (ALFWorld), 76.9% (InterCode), and 58.6% (Wordcraft) on average across 5 training runs. This means 18.7%, 23.1%, and 41.4% of trajectories are discarded respectively. While failed trajectories are not retained for in-context use at test time, failed trajectories are used to improve agent performance via our Exemplar-Curation algorithm (Section 5.3). We do not provide the failed trajectories directly in-context due to the challenges of credit attribution—given a long trajectory with a single incorrect action, it is challenging for the LLM to identify the actions within that trajectory to emulate and the ones to avoid repeating.
>
> **Cost Analysis:** We provide detailed token usage to address your cost-benefit question. Average tokens per request: 5,047/68 input/output tokens (Wordcraft), 5,385/37 (InterCode), 3,706/30 (ALFWorld), with a maximum episode of 8, 20, and 61 requests respectively. More generally, for any task, we require 2 requests per step (reasoning + acting), plus 1 additional request when using planning (see Appendix D6, L550-556). At GPT-4o-mini pricing (0.15/0.6 USD per 1M tokens), cost per episode ranges from 0.004 USD for InterCode to 0.035 USD for ALFWorld. To collect 1000 episodes, maximum API costs range from 4-35 USD.
>
> **Contextualizing the costs of scaling performance along various axes:** In Section 6.3, we demonstrate that our approach can yield substantial cost savings to two common alternatives for scaling model performance: (i) switching to a larger LLM, and (ii) test-time scaling via multiple attempts per task (assuming access to a perfect verifier). On ALFWorld, our full configuration of Traj-Bootstrap + DB-Curation + Exemplar-Curation with 5 parallel databases requires 3,500 × 5 training trajectories. Using GPT-4o-mini, the worst-case cost of constructing the databases is 600 USD, after which per-task `test-time' cost of using GPT-4o-mini is only ~0.034 USD. By comparison, using GPT-4o at test time (without collecting training trajectories) costs ~0.57 USD per task, reaching the same cumulative spend as our approach after ~**1,100** test-time tasks. The alternative baseline of test-time scaling with 3 attempts per task costs a total of ~0.10 USD per task, which exceeds our method's total cost after ~**8,900** test-time tasks. In other words, after a modest up-front investment (of a few hundred dollars of compute) our approach remains **consistently more cost-effective** than either alternative, while also improving test-time agent performance. Consider a hypothetical production agentic service processing a million ALFWorld tasks a day (a very modest industry-scale production throughput). The cost savings **in a single day** of using our method, including all offline dataset construction, would be 530,000 USD vs. GPT-4o, or 60,000 USD vs GPT-4o-mini with 3 trials of test time compute.
>
> > Basically my question is if the improvement is worth the amount of LLM calls? Table 2 shows a 5% increase from 100->3500 training tasks.
>
> You correctly note that a 12-point increase for the ALFWorld task with Traj-Bootstrap comes from the first 100 training tasks, with a 5-point increase in task performance for Traj-Bootstrap when increasing from 100 to 3500 training tasks. We believe this boost is compelling for two reasons: (1) A 5-point improvement from 0.84 to 0.89 success rate represents a 31% reduction in the agent's error rate (from 16% errors to 11% errors)—a substantial improvement. (2) In an appropriately architected production system, this performance boost could come "for free". When running an agent in production, the system could store new experiences in the database, and our results suggest the agent would continue to improve with database size. This represents a key advantage of our approach: performance scaling that naturally accompanies deployment without additional engineering effort.
>
> > Consider also citing some In-Context RL papers such as [1], which also looks at multi-step decision making.
>
> Thank you for the suggestion. Here is an updated related work section we propose to include in our final manuscript:
>
> Our work connects to the emerging area of in-context reinforcement learning, where language models perform sequential decision-making through contextual examples rather than parameter updates. Recent work has explored how transformers can implement RL algorithms in-context: Laskin et al. (2023) demonstrate algorithm distillation for in-context RL, while Lee et al. (2023) investigate transformers' ability to learn from reward trajectory contexts via supervised pretraining. Dai et al. (2024) examine in-context exploration-exploitation strategies, showing that LLMs can balance these competing objectives through prompt design, and Brooks et al. (2023) study in-context policy iteration. While these approaches focus on learning RL algorithms or policies in-context, our work addresses a complementary problem: how to effectively curate and retrieve trajectory examples to maximize in-context learning performance for sequential decision-making tasks.
>
> **References:**
> - Laskin, Michael, et al. "In-context Reinforcement Learning with Algorithm Distillation." ICLR 2023
> - Lee, Jonathan, et al. "Supervised pretraining can learn in-context reinforcement learning." NeurIPS 2023
> - Dai, Zhenwen, et al. "In-context Exploration-Exploitation for Reinforcement Learning." ICLR 2024
> - Brooks, Ethan, et al. "Large language models can implement policy iteration." NeurIPS 2023

---

> > ### Comment · Reviewer_eKr8 · 2025-08-04
> >
> > Thank you for addressing my concerns. I appreciate the thorough cost analysis and also the updated related works. Thinking about it more, I agree that incorporating failed trajectories in-context is a non-trivial problem on its own and would be a lot to tackle in this same paper, given the amount already contributed.
> >
> > I am happy to raise my score.

---

### Decision · Program_Chairs · 2025-09-17

**Decision:**

Accept (poster)

**Comment:**

This paper introduces a trajectory bootstrapping and exemplar curation framework for improving large language model (LLM) agents on sequential decision-making tasks. The central idea is to iteratively collect and curate successful self-generated trajectories, forming a database that can be retrieved and reused as in-context exemplars. Two complementary curation strategies are proposed: DB-level curation, which optimizes the quality of the stored database, and exemplar-level curation, which selects exemplars based on empirical utility. This paper provides a simple and general mechanism for improving LLM agents without requiring domain-specific knowledge. It also provides thorough and systematic experimental evaluation, including extensive ablations and results across multiple benchmarks. Overall, I believe this paper makes a valuable empirical and practical contribution to the field. It proposes a simple yet effective framework for leveraging successful trajectories in sequential decision-making, with strong results supported by rigorous ablation studies. The framing around trajectory bootstrapping and curation is intuitive and potentially impactful for future LLM agent research. I recommend acceptance.